# Structure of a volume-regulated heteromeric LRRC8A/C channel

Sonja Rutz [1], Dawid Deneka [1], Antje Dittmann [2], Marta Sawicka [1] & Raimund Dutzler [1]

Volume-regulated anion channels (VRACs) participate in the cellular response to osmotic swelling. These membrane proteins consist of heteromeric assemblies of LRRC8 subunits, whose compositions determine permeation properties. Although structures of the obligatory LRRC8A, also referred to as SWELL1, have previously defined the architecture of VRACs, the organization of heteromeric channels has remained elusive. Here we have addressed this question by the structural characterization of murine LRRC8A/C channels. Like LRRC8A, these proteins assemble as hexamers. Despite 12 possible arrangements, we find a predominant organization with an A:C ratio of two. In this assembly, four LRRC8A subunits cluster in their preferred conformation observed in homomers, as pairs of closely interacting proteins that stabilize a closed state of the channel. In contrast, the two interacting LRRC8C subunits show a larger flexibility, underlining their role in the destabilization of the tightly packed A subunits, thereby enhancing the activation properties of the protein.

The volume of a vertebrate cell is tightly linked to the osmotic state of its surroundings. While at equilibrium under isotonic conditions, the influx of water in response to a change to a hypotonic environment causes swelling, leading to a dilution of the cytoplasm and in severe cases to bursting. To counteract swelling, cells have developed mechanisms to activate ion and osmolyte efflux pathways in a process called regulatory volume decrease[1,2]. The concomitant efflux of water causes a return of the cell to its original state. Volume-regulated anion channels (VRACs) are important participants in regulatory volume decrease[3,4]. These channels can be activated by an increase of the cell volume and a reduction of the intracellular ionic strength, although the detailed activation mechanism in a physiological context is still poorly understood[5–7]. VRACs are composed of proteins belonging to the conserved LRRC8 family, whose expression is restricted to chordates[8–10]. This family contains five members in humans, termed LRRC8A–E (ref. [11]). All of them share a close sequence relationship and consist of an N-terminal pore domain (PD) followed by a cytoplasmic leucine-rich repeat domain (LRRD)[11]. Although, upon overexpression, several family members can assemble on their own[12,13], in a cellular environment VRACs form obligatory heteromers composed of at least two different homologs[10,14].

In these assemblies, LRRC8A (or SWELL1) constitutes an obligatory subunit, which is essential for the targeting of channels to the plasma membrane[10]. Other subunits determine the substrate selectivity and activation properties of VRACs. Channels containing LRRC8C preferably conduct small inorganic anions and have been identified to play an important role in T-cell regulation[15], whereas the presence of LRRC8D and E extends the range of permeating substrates to osmolytes such as taurine and amino acids[16,17]. Consequently, LRRC8E-containing VRACs in astrocytes have been associated with release of the neurotransmitter glutamate during endemic swelling and stroke, leading to neurotoxic effects[18,19], whereas the channels comprising LRRC8D subunits confer the permeability of platinum compounds, making VRACs an important uptake route for anticancer drugs during chemotherapy[17].

The general architecture of VRACs has been revealed in cryo-EM structures of homomeric LRRC8A[12,20–24], which forms a functional channel with compromised activation properties[12,25,26]. The protein assembles as a hexamer with subunits arranged around an axis of symmetry that defines the ion conduction pore. In these cryo-EM reconstructions, the PDs are generally well-defined, obeying $C6$ or pseudo-$C6$ symmetry, with the LRRDs showing larger conformational heterogeneity[22].

[1]Department of Biochemistry, University of Zurich, Zurich, Switzerland. [2]Functional Genomics Center Zurich, Zurich, Switzerland. ✉e-mail: m.sawicka@gmail.com; dutzler@bioc.uzh.ch

In a major population of particles, adjacent LRRDs have rearranged to maximize interactions leading to subunit pairs forming an asymmetric unit in a *C*3-symmetric protein[12,20,21]. This conformation was found to be stabilized by binding of a synthetic nanobody (sybody), which specifically recognizes the LRRD of the A subunit to inhibit channel activity[27]. A recent structure of a homomeric LRRC8D assembly also displayed a hexameric arrangement of subunits with lower (*C*2) symmetry[13].

In contrast to LRRC8 homomers, our current understanding of heteromeric channels is limited, and restricted to a low-resolution reconstruction of a protein consisting of A and C subunits[12]. Although confirming a hexameric organization as found in LRRC8A channels, the similarity of the subunits has prevented their identification during classification, and the application of *C*3 symmetry, as a measure to improve the density, averaged out conformational differences. Consequently, the disposition of both subunits in hexamers has remained elusive.

To gain insight into the organization of heteromeric VRACs, we engaged in structural studies of channels composed of LRRC8A/C subunits that were obtained by either overexpression or isolation of endogenous protein from native sources. Our study reveals an organization with a single predominant stoichiometry, where A subunits cluster as tightly interacting pairs with a characteristic conformational preference found in homomeric channels, whereas interspersed paired C subunits appear to increase the dynamics of the complex, in line with the proposal that channel activation concomitantly increases the mobility of the LRRDs[27].

## Results

### Distribution of LRRC8 subunits

We first set out to analyze the distribution of subunits in endogenous VRACs, which are expressed under the control of native promotors and assembled by an unperturbed cellular machinery. To this end, we used the sybody Sb$_1$$^{LRRC8A}$ (Sb1), which specifically binds the LRRD of the A subunit with nanomolar affinity[27], for the isolation of endogenous protein from HEK293 cells. Following a tryptic digestion of the purified sample, all five LRRC8 family members were identified by liquid chromatography with tandem mass spectrometry (LC-MS/MS), confirming the described broad expression of subunits in wild-type (WT) cells[10,14] (Extended Data Fig. 1a). Because the observed abundance probably reflects a complex distribution of channel populations with distinct subunit composition, which would prohibit a detailed structural investigation, we turned our attention towards a modified HEK293 cell line carrying genetic knockouts of the B, D and E subunits[10,16] (LRRC8$^{B,D,E−/−}$, generously provided by T. J. Jentsch) in an attempt to reduce sample heterogeneity. However, if binomially distributed in hexameric channels, both subunits could still form 12 distinct assemblies of LRRC8A/C heteromers (Fig. 1a). Similar to HEK293 cells, LRRC8$^{B,D,E−/−}$ cells mediate VRAC currents in response to swelling, which slowly inactivate at positive voltages—this is a hallmark of channels consisting of A and C subunits[14,16] (Extended Data Fig. 1b,c). We then proceeded with affinity purification and MS analysis of endogenous VRACs isolated from WT and LRRC8$^{B,D,E−/−}$ cells, from which we determined LRRC8A/C complex stoichiometries using absolute quantification with two isotopically labeled reference peptides per subunit (Extended Data Fig. 1d). With this approach, we found A and C subunits at a ratio of 1.8:1 in WT cells and at 2.9:1 in LRRC8$^{B,D,E−/−}$ cells (Fig. 1b and Extended Data Fig. 1e). Assuming a hexameric arrangement of channels isolated from LRRC8$^{B,D,E−/−}$ cells and also that our approach has captured all proteins containing LRRC8A, this result reveals a predominance of this obligatory VRAC component in populations presumably containing four to six copies of the subunit. We next attempted to characterize the subunit composition of LRRC8A/C channels produced by heterologous overexpression. To this end, we expressed differentially tagged murine LRRC8A and C constructs either in HEK293S GnTI⁻ cells or in LRRC8$^{−/−}$ cells, where all five LRRC8 subunits have been knocked out[10], and isolated protein by tandem affinity purification to obtain heteromeric

channels that contain at least one copy of each subunit. To probe the variability of the subunit composition, cells were transfected with different ratios of DNA coding for either LRRC8A or LRRC8C subunits. The overexpressed protein was purified and subjected to an analogous MS analysis, as described for endogenous VRACs, that allowed the quantification of subunit ratios. In the case of a transfection of subunits at equimolar ratios, the analysis yielded an A:C ratio of ~2:1 (Fig. 1b and Extended Data Fig. 1e). Although this ratio can be slightly perturbed upon the transfection of LRRC8C-DNA at three times higher concentration compared to LRRC8A-DNA, the resulting A:C subunit ratio of 1.8:1 emphasizes that even at a large excess of the former, the LRRC8A subunits prevail (Fig. 1b and Extended Data Fig. 1e). Together, our results hint at a dominating distribution of A subunits in heteromeric LRRC8A/C channels, which contrasts with a recent proposal that LRRC8A is a minor component of VRACs[28].

### Structural properties of LRRC8C homomers

In the next step, we engaged in the structural characterization of the building blocks of the VRAC heteromers. By combining data from cryo-EM and X-ray crystallography, previous studies have revealed the structural properties of LRRC8A[12,20–22], which assembles as a hexamer, exhibiting characteristic features of an ion channel. Here we investigated whether we would find similar properties for homomeric assemblies consisting of LRRC8C subunits. We thus expressed full-length LRRC8C and its isolated LRRD (LRRC8C$^{LRRD}$) and purified both constructs for further characterization. As for the LRRD of the A subunit, LRRC8C$^{LRRD}$ is a monomeric protein in solution. Its structure determined by X-ray crystallography at 3.1 Å (Extended Data Fig. 2) shows features similar to those of the A domain, with both horseshoe-shaped proteins superimposing with a root-mean-square deviation (r.m.s.d.) of 1.4 Å (Fig. 2a). The domains share a sequence identity of 56%, consist of the same number of repeats and do not contain insertions in loop regions. We then set out to characterize the full-length protein and collected cryo-EM data for an LRRC8C homomer from three independent preparations. Unexpectedly, the two-dimensional (2D) classes from these datasets showed in all cases a heptameric assembly, which was confirmed by 3D reconstruction (Extended Data Fig. 3). Despite the large size of the combined dataset, the non-symmetrized reconstruction of the full-length protein did not reach high resolution, probably due to the intrinsic mobility of the complex. After application of *C*7 symmetry, we were able to obtain a structure extending to 4.6 Å for the full-length protein and 4.1 Å for the PD (Extended Data Fig. 3b and Table 1). At low contour, the map displays an envelope for the entire protein, and at higher threshold, where the density of the more mobile LRRDs has largely disappeared, it defines the structure of its membrane-inserted portion (Fig. 2b,c). This map allowed a molecular interpretation with subunits consisting of the structure of the PD determined by cryo-EM and the X-ray structure of the LRRD to obtain a symmetric channel with a pore radius of 6 Å at its extracellular constriction (Fig. 2d–f). In this assembly, close subunit interactions are restricted to the extracellular part, whereas contacts within the remainder of the protein are scarce (Fig. 2e). With respect to its subunit organization and pore size, the PD of the homomeric LRRC8C closely resembles the heptameric pannexin channel, which is known to conduct large substrates, including adenosine triphosphate[29] (ATP; Fig. 2f). The reported lack of activity is thus probably a consequence of its cellular distribution, as the subunit, when expressed on its own, is not targeted to the plasma membrane[10,26], whereas functional LRRC8C channels have been obtained in a chimera containing a disordered loop of LRRC8A that promotes expression at the cell surface[26]. This construct was recently shown to assemble as a heptamer[30].

### Structure of LRRC8A/C channels in complex with Sb1

To gain insight into the structural properties of the LRRC8A/C channels, we overexpressed the protein in large suspension cultures of HEK293

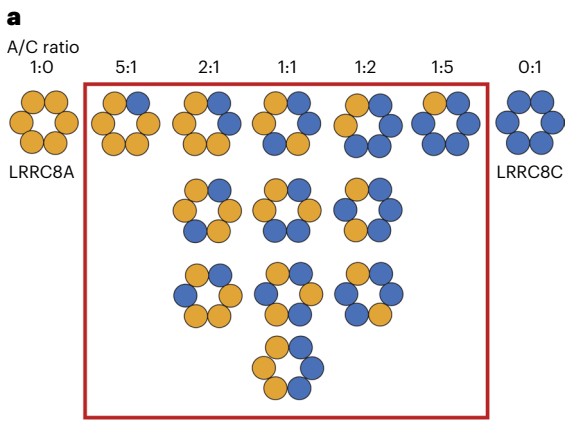

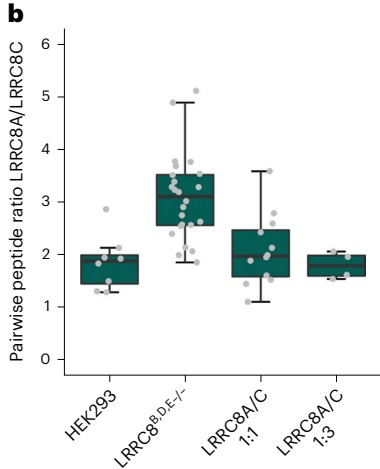

**Fig. 1 | MS analysis of LRRC8A/C. a**, Schematic of the possible distribution of subunits in heteromeric LRRC8A/C channels. Subunits in hexameric channels are represented by circles of different colors. The red box encloses all channel populations containing at least one copy of each subunit. **b**, Determination of the ratio of LRRC8A to LRRC8C in isolated complexes using LC-MS/MS. Pairwise ratios of LRRC8A peptides relative to LRRC8C peptides were either generated for endogenous LRRC8 protein obtained from HEK293 or LRRC8$^{B,D,E-/-}$ cells or overexpressed protein expressed after transfection with different ratios of DNA coding for LRRC8A and C subunits (that is, at A:C DNA ratios of 1:1 and 1:3). Absolute peptide amounts calculated by spiking each sample with known amounts of stable isotope-labeled peptides were used for ratio determination. Boxplots cover the first and third quartiles from bottom to top, and the whiskers extend to largest/smallest value but no further than 1.5 × IQR (interquartile range). The median ratio is indicated by a black solid line.

**Fig. 2 | LRRC8C structure. a**, Superposition of the X-ray structures of the LRRDs of LRRC8C and A. **b,c**, Cryo-EM density of full-length LRRC8C at 4.6 Å (**b**) and of the masked PD at 4.1 Å (**c**). The view is from within the membrane. **d**, Structure of the LRRC8C heptamer with structural regions indicated. In **a** and **d**, proteins are shown as ribbons. **e**, Molecular surface of the LRRC8C heptamer with the relative orientation compared to the membrane view (center) indicated. In **d** and **e**, membrane boundaries are shown as lines. **f**, Pore radius in the filter regions of the LRRC8C heptamer and the heptameric Pannexin 1 (PDB 6WBF) calculated by HOLE[41].

**Table 1 | Cryo-EM data collection, refinement and validation statistics**

| | LRRC8C (EMD-15835, PDB 8B4O) | LRRC8A/C[1:1]/Sb1 (EMD-15836, PDB 8B41) | LRRC8A/C (EMD-15837, PDB 8B42) |
|---|---|---|---|
| **Data collection and processing** | | | |
| Magnification | 130,000 | 130,000 | 130,000 |
| Voltage (kV) | 300 | 300 | 300 |
| Electron exposure (e⁻/Å²) | 59–67 | 67 | 67 |
| Defocus range (µm) | −2.4 to −1.0 | −2.4 to −1.0 | −2.4 to −1.0 |
| Pixel size (Å)ª | 0.651 (0.3255) | 0.651 (0.3255) | 0.651 (0.3255) |
| Symmetry imposed | C7 | C1 | C1 |
| Initial micrographs (no.) | 33,576 | 26,442 | 14,160 |
| Initial particle images (no.) | 2,016,749 | 1,813,389 | 1,205,995 |
| Final particle images (no.) | 137,432 | 329,716 | 119,006 |
| Map resolution FL, PD (Å) | 4.6, 4.1 | 3.8, 3.3 | 6.6 |
| FSC threshold 0.143 | | | |
| Map resolution range (Å) | 4.0–7.2 | 2.5–8.0 | 4.8–13.5 |
| **Refinement** | | | |
| Initial model used (PDB code) | 7P5V | 7P5V | 7P5V |
| Model resolution (Å) | 7.2 | 3.9 | 8.3 |
| FSC threshold 0.5 | | | |
| Map sharpening B factor (Å²) | −160 | −103 | −153 |
| Model composition | | | |
| Nonhydrogen atoms | 39,466 | 31,259 | 31,849 |
| Protein residues | 4,844 | 3,805 | 3,866 |
| B factors (Å²) | | | |
| Protein | 182.0 | 133.5 | 521.6 |
| R.m.s. deviations | | | |
| Bond lengths (Å) | 0.002 | 0.002 | 0.002 |
| Bond angles (°) | 0.495 | 0.481 | 0.548 |
| Validation | | | |
| MolProbity score | 2.36 | 2.24 | 2.53 |
| Clashscore | 11.76 | 9.39 | 14.92 |
| Poor rotamers (%) | 4.29 | 4.69 | |
| Ramachandran plot | | | |
| Favored (%) | 95.74 | 96.46 | 96.16 |
| Allowed (%) | 4.24 | 3.54 | 3.8 |
| Disallowed (%) | 0.02 | 0.0 | 0.0 |

ªValues in parentheses indicate the pixel size in super-resolution. FL refers to the full-length channel, PD to its pore domain.

cells transfected with constructs coding for the A and C subunits at two distinct ratios (LRRC8A:C ratios of 1:1 and 1:3), isolated the channels by tandem affinity purification and added the sybody Sb1 at a 1.5 molar excess before vitrification. For both preparations, we have collected large datasets by cryo-EM and proceeded with 2D classification and 3D reconstruction. Of the two datasets, only the one collected from channels obtained from a transfection with an equimolar ratio of DNA (LRRC8A/C[1:1]/Sb1), showing an A-to-C stoichiometry of 2:1, permitted reconstruction at high resolution (Extended Data Figs. 4–6 and Table 1). In contrast, the particles in a sample obtained from transfection with a 1:3 ratio of A to C subunits (LRRC8A/C[1:3]/Sb1) resulting in a

subunit ratio of 1.8:1 are less well ordered, which has complicated alignment, and thus compromised the obtained resolution (Extended Data Fig. 7a–d and Table 2). In agreement with previous studies, the channels in both datasets are hexameric, and we did not spot any heterogeneity with respect to their oligomeric state. To compare the structural properties of overexpressed samples with endogenous VRACs, we also purified channels from LRRC8[B,D,E−/−] cells using a column containing immobilized Sb1 for affinity chromatography. The eluted endogenous LRRC8A/C[endog]/Sb1 complex was frozen on carbon-supported grids and used for cryo-EM data collection. The poor yield of this preparation and the consequent low particle density on the grids together with the potential heterogeneity of particles suggested by our MS analysis (Fig. 1b and Extended Data Fig. 1e) prevented the unambiguous alignment of distinct subunits in the VRAC heteromers and thus restricted our analysis to general attributes derived from a reconstruction at low resolution (Extended Data Fig. 7e–i and Table 2). Despite these limitations, we found a structure carrying characteristic features observed for the overexpressed samples (Extended Data Fig. 7f,i).

In our study, the structure of the LRRC8A/C[1:1]/Sb1 complex has defined the properties of a heteromeric VRAC at high resolution (Fig. 3a, Extended Data Figs. 4–6, Table 1 and Supplementary Video 1). Although we expected to observe a heterogeneous population of channels, following 3D classification we found a hexameric protein with nearly uniform subunit distribution in a single predominant conformation (Extended Data Fig. 4b). In this assembly, four adjoining subunits including their cytoplasmic LRRDs are well-defined, with the density of bound Sb1 distinguishing them as LRRC8A chains, whereas the density of the LRRDs of the two remaining subunits is absent (Fig. 3a). The four A subunits in the hexamer (denoted A1–A4, Fig. 3a,b) are organized as pairs with mutual tight interactions between their LRRDs, as initially observed in the homomeric LRRC8A complex[12,27] (Fig. 3b,c). In contrast, the two pairwise interacting C subunits (denoted C1 and C2) are more dynamic. Whereas the extracellular portion of the PDs of both C subunits, consisting of the extracellular subdomains (ESDs) and the membrane-inserted segments (TMs), are well-defined and show structural hallmarks of this paralog, the cytoplasmic subdomain (CSD) of the C1 subunit located at the A4/C1 interface is defined poorly, and that of C2 located at the C2/A1 interface is not resolved (Fig. 3a). Additionally, both LRRDs are mobile and thus not visible in the cryo-EM density (Fig. 3a). The observed organization reflects the properties of the respective homomeric structures exhibiting extended interactions between LRRC8A subunits, whereas the C subunits are less well packed. Within the pore domain, the ESDs obey pseudo-six-fold symmetry, which is also largely maintained for the TMs and CSDs of the A subunits, whereas the TM of C1 has undergone a slight outward movement away from the pore axis that can be described by a 3° rigid body rotation around an axis placed at the border between ESD and TM, reflecting the apparent deterioration of the interactions at A/C, C/C and C/A interfaces (Fig. 3d). In the structure of the A subunit in complex with Sb1 (ref. [27]), the sybody has led to a rigidification of the domain structure, which is also observed in the LRRC8A/C heteromer (Fig. 3c). In this arrangement of the A subunits, the conformation of the C subunits observed in the heptameric LRRC8C structure would lead to clashes, which are pronounced between the LRRDs of C2 and A1 (Fig. 3e), thus forcing them into a different conformation. However, instead of adopting an A-like domain arrangement, which would allow their accommodation in the restricted space of the hexameric protein, the LRRDs of the C domains have become mobile and are thus not defined in the density of the heteromeric channel.

### Structure of LRRC8A/C channels in the absence of Sb1

Because the binding of Sb1 rigidifies the LRRDs of the four A subunits in the observed conformation, thereby restricting the accessible space of the corresponding domains of the two LRRC8C subunits, we continued to characterize structures of LRRC8A/C in the absence of the sybody.

**Table 2 | Cryo-EM data collection statistics of low-resolution datasets**

| | LRRC8A/C[1:3]/Sb1 (EMDB-15838) | LRRC8A/C[endog]/Sb1 (EMDB-15839) | LRRC8A[SAM] (EMDB-15840) | LRRC8A[SAM]/C (EMDB-15841) |
|---|---|---|---|---|
| **Data collection and processing** | | | | |
| Magnification | 130,000 | 130,000 | 160,000 | 130,000 |
| Voltage (kV) | 300 | 300 | 300 | 300 |
| Electron exposure (e⁻/Å²) | 67 | 59 | 56 | 59 |
| Defocus range (µm) | −2.4 to −1.0 | −2.4 to −1.0 | −2.5 to −0.8 | −2.4 to −1.0 |
| Pixel size (Å)[a] | 0.651 (0.3255) | 0.651 (0.3255) | 1.31 | 0.651 (0.3255) |
| Symmetry imposed | C1 | C1 | C3 | C1 |
| Initial micrographs (no.) | 33,672 | 47,988 | 1,677 | 24,560 |
| Initial particle images (no.) | 1,930,456 | 2,589,543 | 206,490 | 2,451,262 |
| Final particle images (no.) | 98,883 | 376,175 | 41,806 | 49,929 |
| Map resolution (Å) | 9.5 | 18.2 | 6.9 | 7.8 |
| FSC threshold 0.143 | | | | |
| Map sharpening B factor (Å²) | −847 | N/A | −270 | −413 |
| **Refinement** | N/A | N/A | N/A | N/A |

[a]Values in parentheses indicate the pixel size in super-resolution.

For that purpose, we followed two different strategies, one involving the labeling of the PD of the A subunit to facilitate its identification and particle alignment and a second, the classification of LRRC8A/C channels without any labeling. For the first approach, we generated a construct where we fused the 57-residue-long SAM (sterile alpha motif) domain of human tumor suppressor p73 to the truncated first extracellular loop of LRRC8A to create the construct LRRC8A[SAM] (Extended Data Fig. 8a). The replacement of the mobile loop was well tolerated, and a dataset of homomeric LRRC8A[SAM] showed a channel with conformational properties similar to those observed in the structure of the unlabeled A subunit where the SAM domain is clearly recognizable in a low-resolution reconstruction, distinguishing it as a proper fiducial marker (Extended Data Fig. 8b and Table 2). The heteromeric channel obtained from a 1:1 ratio of transfected constructs (LRRC8A[SAM]/C) is functional (Extended Data Fig. 8c,d) and contains a similar 2:1 ratio of A:C subunits as WT LRRC8A/C (Extended Data Fig. 8e,f). A cryo-EM dataset of this sample revealed a larger heterogeneity than observed in the LRRC8A/C/Sb1 complex with two prominent channel populations, one equivalent to the LRRC8A/C[1:1]/Sb1 complex and a second showing an altered arrangement where both C subunits are placed on opposite sides of the hexamer (Extended Data Fig. 8g–k). Both populations are averaged in a reconstruction at 7.8 Å (Extended Data Fig. 8k). The altered subunit disposition in this channel population is probably a consequence of the fused SAM domain, which appears to mildly perturb the interaction between LRRC8A subunits, leading to the dissociation of contacts at the loose interface. Although these properties illustrate that even a considerable modification of the expression construct might affect the channel assembly, the preserved 2:1 A-to-C stoichiometry and the pairwise organization of tightly interacting LRRC8A subunits further support their role as building blocks in heteromeric VRACs.

An arrangement closely resembling the LRRC8A/C[1:1]/Sb1 complex was observed in a dataset of LRRC8A/C obtained in the absence of Sb1. In the 3D reconstruction generated from this dataset, we find a consecutive arrangement of four well-defined subunits that are organized as tightly interacting pairs and two less well-defined subunits (Fig. 4, Extended Data Fig. 9, Table 1 and Supplementary Video 2). In the latter, additional density at the level of the LRRDs can be attributed to the subunit occupying the C1 position, suggesting that the absence of Sb1 would allow for a better integration of the C subunits in the hexameric protein (Fig. 4a). In this structure, the A subunits are readily identified by their characteristic pairing observed in previous structures (Fig. 3a).

However, the LRRDs have rearranged compared to the interactions in the LRRC8A/C/Sb1 complex (by rigid body rotations of 16°, 9°, 11° and 10° for the respective positions A1–A4), leading to a weakening of the tight interface and the creation of a gap between interacting domain pairs (Fig. 4b,c and Supplementary Video 3). The described movements of the LRRDs of the four A subunits have expanded the accessible space for the respective regions of the adjacent C subunits, as manifested in the emergence of density of the LRRD of the subunit in the C1 position and of the CSD of the less-well-defined C2 position (Fig. 4a). Remarkably, the relative orientation of the LRRD in the C1 position is distinct from the conformations observed for the A subunits. It instead resembles the arrangement in the LRRC8C heptamer, except for a rigid body rotation by 17° away from the pore axis around a pivot that is located at the interface to the PD (Fig. 4c,d). A similar LRRD conformation at the C2 position would result in a clash with the contacted A1 position, requiring a moderate rearrangement that increases the domain mobility, as reflected in its absent density. Together, our results emphasize the distinct conformational preferences of the A and C subunits, defined in the datasets of the respective homomers, as determinants of their properties in heteromeric channels.

**The anion selectivity filter**

In contrast to its intracellular parts, the TMs and ESDs of the C subunits in the LRRC8A/C[1:1]/Sb1 dataset are well defined and provide detailed insight into the structural properties of a heteromeric VRAC (Fig. 5a). These are particularly informative at the level of the ESDs, which form the constricting part of the channel, resembling a selectivity filter (Fig. 5b and Extended Data Fig. 6c). In the extracellular half of the PD, the pseudo-symmetry-related subunits are found in a similar arrangement as observed in the homomeric LRRC8A (Fig. 5c). In this part of the channel, the surface area buried between contacting subunits, ranging between 2,200 and 2,500 Å², is of comparable size in all interfaces (Fig. 5b). Still, subunit-specific differences, such as the replacement of residues engaged in a salt bridge in A/A interfaces (between His104 and Asp110) by uncharged polar residues (Gln106 and Asn112) might modulate the strength of the interaction (Fig. 5d,e). At the constriction, the A subunits contain an arginine (Arg103), which determines the high anion-over-cation selectivity of the channel[12] (Fig. 5e). In the case of the C subunits, this arginine is replaced by a leucine (Leu105) whose lower side chain volume increases the pore diameter at the constriction (Fig. 5d,f), thus probably accounting for the increased

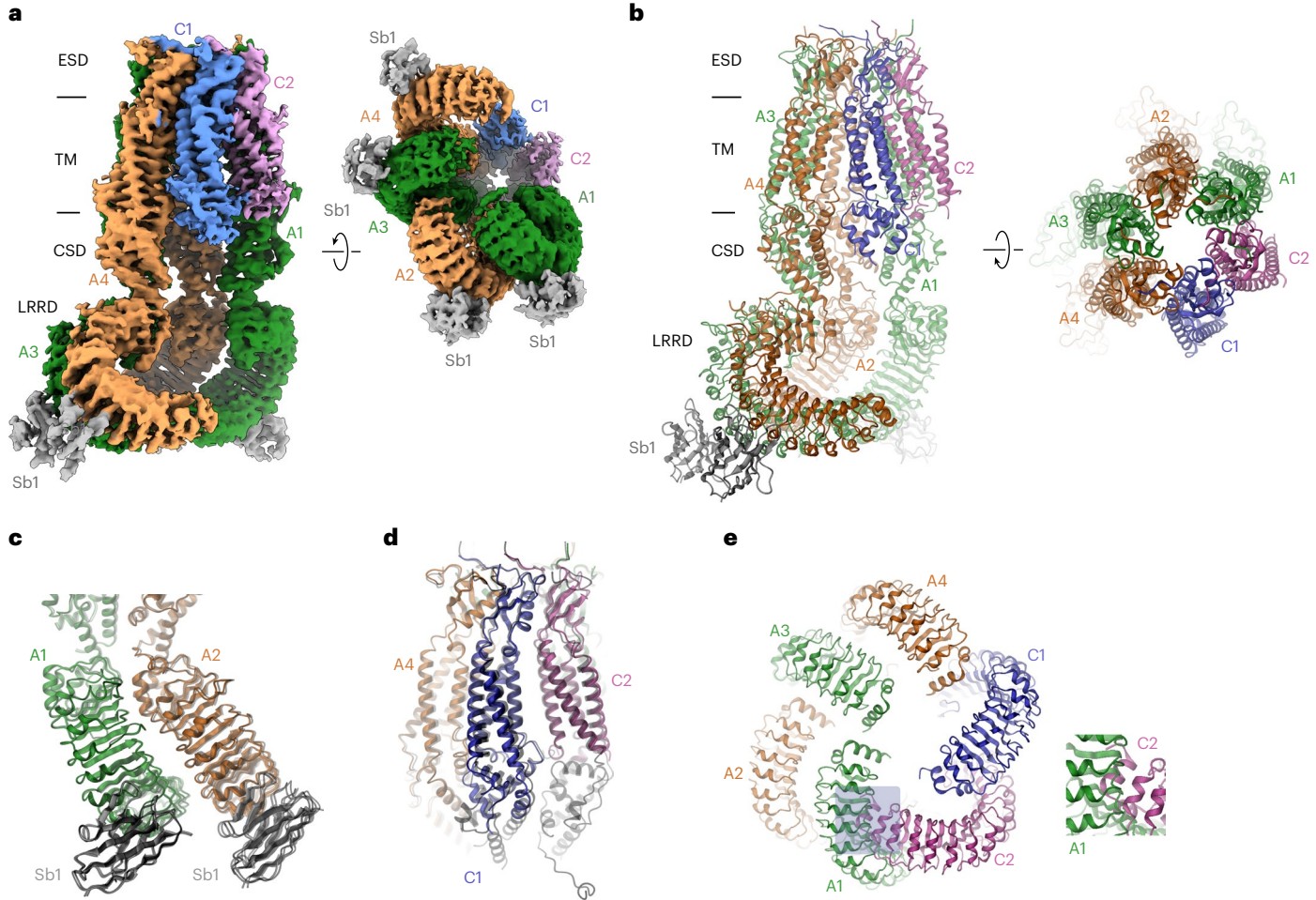

**Fig. 3 | Structure of the LRRC8A/C[1:1]/Sb1 complex. a**, Cryo-EM density of the entire LRRC8A/C[1:1]/Sb1 complex at an overall resolution of 3.8 Å. The views are from within the membrane (left) and from the cytoplasm (right). **b**, Ribbon representation of the LRRC8A/C/Sb1 complex viewed from within the membrane (left) and from the extracellular side (right). In **a** and **b**, membrane boundaries, structural elements and subunit positions are labeled. **c**, Structure of the tightly interacting LRRD pair of A subunits with bound Sb1. **d**, View of the PD. In **c** and **d**,

the superimposed structure of the equivalent units of the LRRC8A/Sb1 complex (PDB 7P5V) is shown for comparison (black, transparent). **e**, Arrangement of LRRDs in the LRRC8A/C/Sb1 complex. The A domains are shown in the observed conformation. The C domains modeled in the conformation observed in the LRRC8C structure would lead to steric clashes that are pronounced in the C2/A2 interface (boxed region and enlarged inset). In all panels, subunits are shown in unique colors, and bound sybodies are in gray.

single channel conductance of the A/C complexes compared to the A homomers found in a previous study[14]. The described difference also alters the polar properties of the filter by introducing a hydrophobic segment into a ring of basic residues.

## Discussion

Our study provides detailed insight into the previously unknown organization of heteromeric VRACs. In a cellular environment, these proteins consist of the obligatory LRRC8A subunit and at least another member of the LRRC8 family, to form functional channels with distinct composition-dependent properties[10,16,17]. Their hexameric architecture and the possibility to assemble proteins from five different homologs leads to a vast number of possible arrangements. A large heterogeneity of heteromers would thus be expected in the case where all subunits interact with similar affinity and their assembly were governed by thermodynamics[23]. In such a scenario, the distribution of distinct oligomers would exclusively depend on the concentration of expressed subunits within a cell, and their relative disposition in the channel would be random. To limit the number of possible assemblies, we thus focused on heteromers formed by the protein chains LRRC8A and C, which in a hexameric channel could form up to 12 distinct assemblies (Fig. 1a). By employing absolute quantification of proteins by MS, we found

a robust 2:1 ratio of A-to-C homologs in samples purified from cells transfected with equimolar amounts of DNA, which is also reflected in the structural properties of the sample (Fig. 3). The higher 3:1 ratio of A-to-C subunits, observed in endogenous channels purified from cells where other homologs were genetically knocked out, reflects a heterogeneous distribution where channels with a 2:1 subunit ratio would constitute a major population. A 5:1 ratio of A-to-C subunits was reported in a recent structural study of a heteromeric complex containing a genetically modified fusion construct of LRRC8A[31], resembling the approach taken with the LRRC8A[SAM] fusion used here. The nature of this discrepancy is currently unclear and could be either a consequence of the used construct or related to the different expression host. The observed abundance of A subunits in heteromeric channels contradicts a previous proposal suggesting that LRRC8A might be a minor component of VRACs[28]. Assuming an unbiased distribution of subunits in channels with an A-to-C ratio of 2:1, we would still expect to find three distinct assemblies (Fig. 1a). In contrast, we find a single distribution with A and C subunits segregating into clusters, suggesting that the affinity between homomers prevails. Differences in the conformational properties of distinct subunits underlying their observed clustering in heteromeric assemblies can already be appreciated in structures of homomers. Homomeric LRRC8A channels are distinguished by their

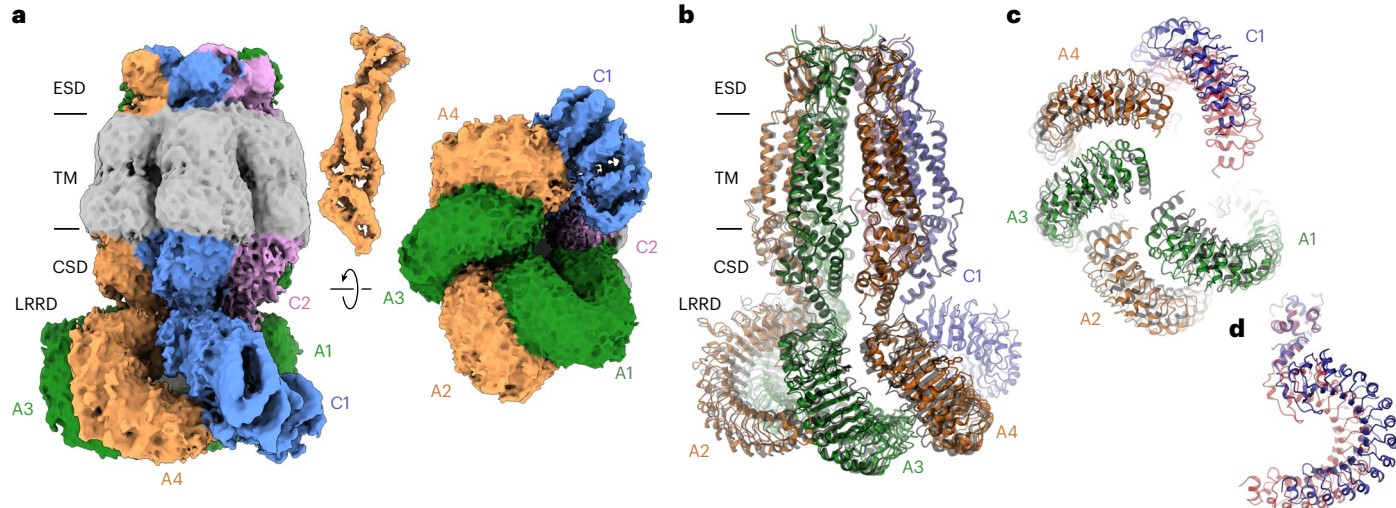

**Fig. 4 | Structure of LRRC8A/C in the absence of Sb1. a**, Cryo-EM density of the entire LRRC8A/C complex with a masked protein region (excluding the detergent belt) at an overall resolution of 6.6 Å. The view is from within the membrane (left), with the inset (top) showing the density of the PD of the A4 subunit at higher contour, and from the cytoplasm (right). **b**, Ribbon representation of the LRRC8A/C complex viewed from within the membrane. **c**, LRRD conformations viewed from the cytoplasm. **d**, LRRD of LRRC8C in the C1 position. In **b** and **c**, the superimposed structure of the equivalent units of the LRRC8A/C/Sb1 complex is shown for comparison (black, transparent). In **c** and **d**, the structure of a single superimposed subunit of the LRRC8C homomer is shown in red for comparison. In **a** and **b**, membrane boundaries, structural elements and subunit positions are labeled.

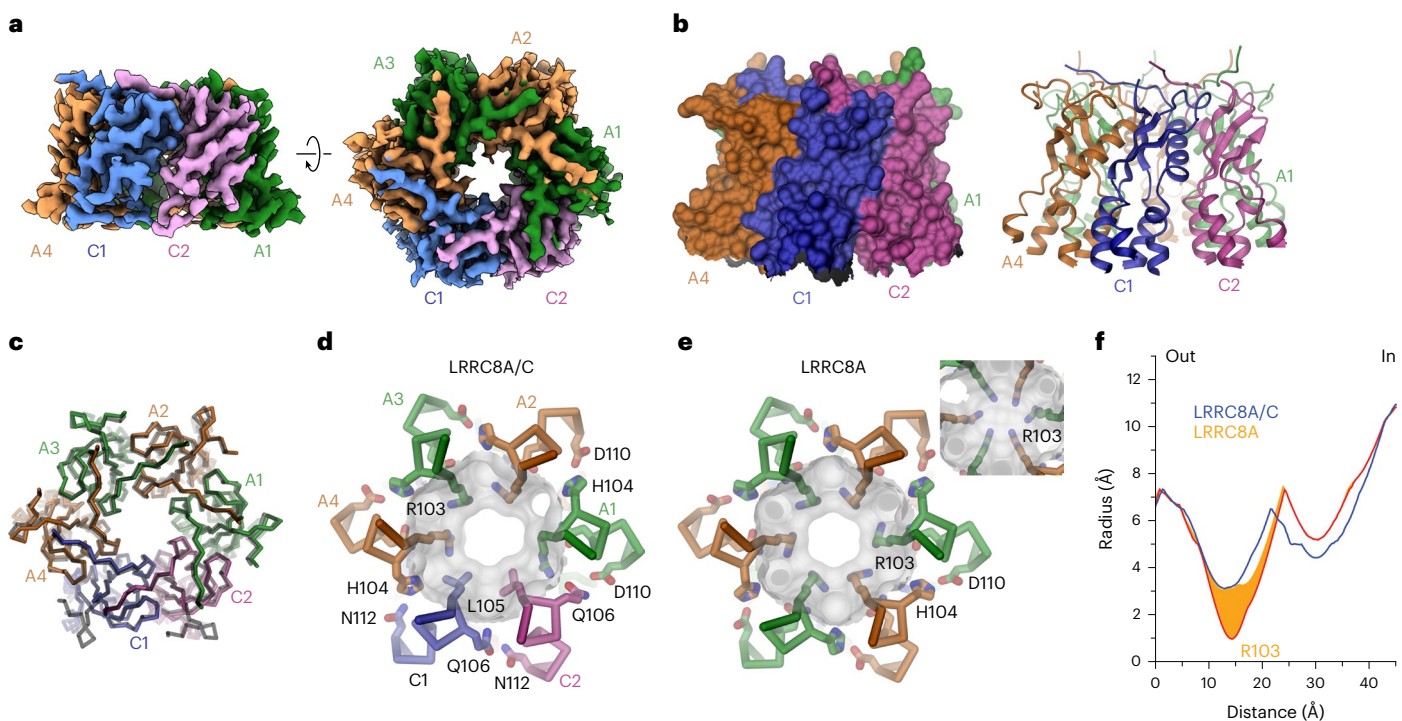

**Fig. 5 | LRRC8A/C selectivity filter. a**, Cryo-EM density of the masked extracellular part of the PD containing the ESDs and the extracellular part of the TMs at a resolution of 3.1 Å. The views are parallel to the membrane (left) and from the extracellular side (right). **b**, Molecular surface (left) and ribbon representation (right) of the ESDs and the adjacent region of the TMs viewed parallel to the membrane. **c**, Superposition of Cα traces of the ESDs of LRRC8A/C and the equivalent region of LRRC8A (black transparent). **d,e**, Comparison of the pore constrictions of LRRC8A/C (**d**) and LRRC8A (**e**).

The protein is shown as a Cα trace, with selected side chains displayed as sticks. The molecular surface is shown superimposed. Inset in **e**: the LRRC8A pore with altered side chain conformation of Arg 103 altering the pore diameter. **f**, Pore radius in the filter regions of the LRRC8A/C heteromer and homomeric LRRC8A (PDB 7P5V) calculated by HOLE[41]. The orange area shows the distribution of the pore radii of LRRC8A in dependence on the Arg 103 conformation (with the red line corresponding to the minimal diameter from the conformation shown in **e**, inset).

compact oligomeric arrangement leading to the formation of tightly interacting subunit pairs where the comparably mobile LRRDs have rearranged to maximize interactions[12,20,21]. It is thus not surprising to also find interacting LRRC8A pairs as invariant building blocks in heteromeric channels. The observed tight interaction of A subunits is consistent with the poor activation properties of LRRC8A homomers[12,14,25,26],

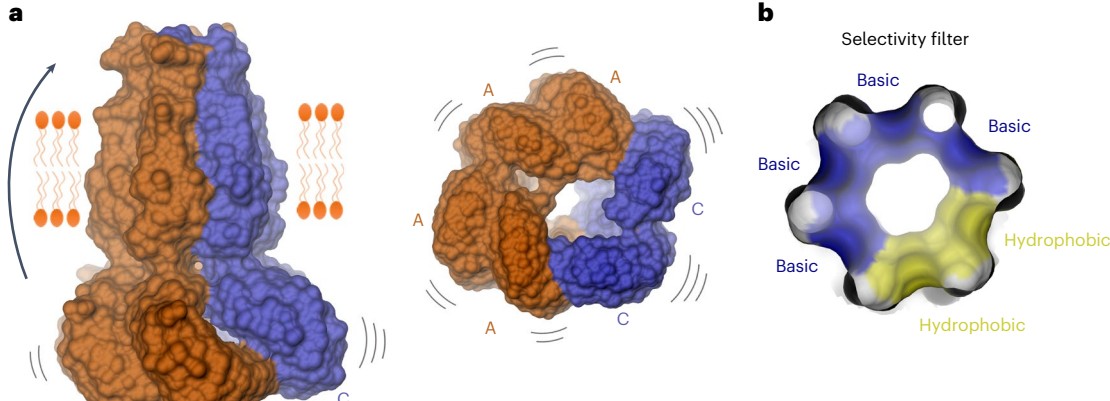

**Fig. 6 | Features of a heteromeric LRRC8 channel. a**, Schematic of a heteromeric LRRC8A/C channel viewed from within the membrane (left) and from the cytoplasm (right). The mismatch of preferred conformations of the LRRC8A (orange) and LRRC8C (blue) subunits compromises the tight interaction between the LRRC8A subunits, with the increasing dynamics enhancing channel activity. **b**, View of the selectivity filter constituted by the ESDs of an LRRC8A/C channel from the extracellular side. The molecular surface of the filter is colored according to the physico-chemical properties of the contacting residues.

suggesting that they stabilize a closed state of the ion conduction pore. In contrast, there are fewer contacts in the case of LRRC8C homomers, which have formed a larger heptameric assembly (Fig. 2). In this case, the LRRDs show increased mobility compared to the A subunits, and we did not find any indication of tight subunit interactions. Similarly, the interaction interface in the PD is reduced and restricted to contacts between the ESDs (Fig. 2). Considering the observed symmetry mismatch, the incorporation of C subunits and potentially also other LRRC8 homologs into heteromeric channels would perturb the tight interactions found in A homomers and thus destabilize the closed state and improve the activation properties (Fig. 6a). This is illustrated in the observed weaker density of C subunits in the LRRC8A/C[1:1]/Sb1 complex, which is pronounced at the level of their intracellular components (Fig. 3a) and the disruption of LRRD contacts in tightly interacting subunit pairs leading to a conformational change in the structure of an LRRC8A/C channel in the absence of Sb1 (Fig. 4). This mechanism is consistent with the previously proposed role of potentiating sybodies to perturb LRRD packing[27], as well as the observed correlation between increased LRRD mobility and activation[32,33]. Although, in combination with earlier studies[27,32,33], our current data strongly support the notion of LRRDs to regulate channel activity by coupling to a physical gate, the exact location of this gate and the nature of the coupling mechanism remains poorly understood[23]. Previous studies have assigned a role of the ESDs in voltage-dependent inactivation[34] and suggested the N termini, which project into the pore, to be a major constituent of the gate[35]. In contrast to the ordered N termini in the structure of LRRC8D[13], the equivalent regions of the A and C subunits appear mobile and are thus not defined in the structures of homo- and heteromeric channels. The mechanism by which they might contribute to the inhibition of ion conduction is thus still unclear. A recent study has proposed a role of pore-lining lipids in channel gating based on residual density at the extracellular part of the TM domain[31]. Although similar weak density is found in the LRRC8A/C[1:1]/Sb1 complex, it is not sufficiently detailed to warrant such a conclusion (Extended Data Fig. 10). A potential role of lipids in VRAC gating thus requires further investigation.

The observed organization of subunits also affects the properties of the ESDs constituting the anion selectivity filter, where the substitution of a constricting arginine in LRRC8A by a smaller leucine in LRRC8C has increased the pore diameter and introduced a hydrophobic segment in the filter that resembles the amphiphilic properties of equivalent regions found in unrelated chloride channels and transporters[36–38] (Figs. 5d,f and 6b). Knowledge of the detailed filter architecture could be exploited in the design of potent and specific pore blockers binding to this region, as a strategy against diabetes and other VRAC-related diseases[39,40]. An increased pore diameter can also be expected for heteromeric channels containing D and E subunits[13,16,17], although it is currently unclear whether in a hexameric organization this increase would be sufficiently large to account for the pronounced properties of these channels that renders them permeable to larger substrates such as amino acid osmolytes and anticancer drugs[17]. In that respect, the larger heptameric assembly of LRRC8C homomers described in this study is noteworthy, as it would allow for a further increase of the pore radius, although there is so far no evidence for the existence of heteromeric channels with larger oligomeric organization in a physiological context. It will thus be important in future studies to examine the assembly of heteromeric VRACs composed of different family members and containing more than two distinct subunits to better understand their versatile functional properties and unique activation mechanism. However, also in channels with alternate subunit composition, we expect the general rules defined in the present study to apply.

## Online content

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

## Methods

### Expression constructs and cloning

All constructs were generated using FX-cloning and FX-compatible vectors[42]. The constructs encompassing murine LRRC8A, LRRC8C and LRRC8C[LRRD] were obtained from previous studies[12,27]. For generation of the LRRC8A[SAM] fusion construct, 57 residues of the SAM domain from human p73 (PDB 1DXS) were inserted into a truncated extracellular loop region of murine LRRC8A. The sequence coding for the rigid core of the SAM domain (encompassing residues 8–64) with flanking Sap1 recognition sites at both termini was obtained from GenScript (GCTCTTCTTCCA GCCTGGTGTCCTTCCTGACCGGACTGGGATGCCCCA ACTGTATCGAGT ACTTTACATCTCAGGGCCTGCAGAGCATCTATCACCTGCAGAATCTGAC CATCGAGGACCTGGGCGCCCTGAAGATCCCTGAGCAGTACCGGAT GACAATCTGGAGAGGCCTGCAGGATCTGACGTGAAGAGC). A linearized plasmid of a murine LRRC8A expression construct lacking the sequence coding for residues 68–92 and containing SapI sides on both ends was generated by polymerase chain reaction (PCR; forward primer, TATAGCTCTTCTACGGGCCCTACCG; reverse primer, TATAGCTCTTCAG GAGTCTTGGC AGCTGTCCTTGGTG). Subsequently, the SAM domain sequence was inserted into the plasmid using FX cloning. The N terminus of this SAM domain contains an insertion of a Ser, and the C terminus of a Thr residue. Additionally, Asn 67 of LRRC8A was mutated to a Gln. For protein expression of homomers, full-length LRRC8C or LRRC8A[SAM] was cloned into a pcDX vector containing a C-terminal Rhinovirus 3C protease-cleavable linker (3C cleavage site) followed by Venus[43] (for LRRC8C) or mCherry[44] (for LRRC8A[SAM]), a myc-tag and streptavidin-binding peptide[45] (SBP, pcDXc3VMS or pcDXc3ChMS). LRRC8A and LRRC8A[SAM] were cloned into pcDX vectors containing a 3C cleavage site followed by mScarletI[46] and a His[10]tag (pcDXc3SH) for co-expression experiments with LRRC8C. For electrophysiology, LRRC8A, LRRC8A[SAM] and LRRC8C were cloned into pcDX vectors containing a 3C cleavage site followed by a myc tag and an SBP tag (pcDXc3MS). The Venus-only construct, which was co-expressed with constructs used for patch-clamp recordings, contained the Venus gene, followed by a myc tag and a SBP tag. LRRC8C[LRRD], consisting of residues 395–803, was cloned into a pcDX vector containing an N-terminal SBP and a myc-tag followed by a 3C cleavage site (pcDXn3MS). For periplasmic expression of Sb1 in bacteria, the sybody was cloned into an arabinose-inducible vector containing an N-terminal pelB signal sequence and a His[10]-tag followed by a 3C cleavage site (pBXnPH3)[47]. For purification of endogenous LRRC8 protein from HEK293 cells, Sb1 was expressed in the cytoplasm of mammalian cells using a pcDX vector containing a C-terminal C3 cleavage site followed by a myc-tag and a SBP tag (pcDXc3MS).

### Cell culture

HEK293S GnTI⁻ (CRL-3022) and HEK293T (CRL-1573) cells were obtained from ATCC. HEK293 LRRC8⁻/⁻ (LRRC8⁻/⁻) and HEK293 LRRC8[B,D,E–/–] cells (LRRC8[B,D,E–/–])[10,16] were kindly provided by T. J. Jentsch. All four cell lines were adapted to suspension cultures and grown in HyCell HyClone TransFx-H medium (Cytivia) supplemented with 1% FBS, 4 mM L-glutamine, 100 U ml⁻¹ penicillin-streptomycin and 1.5 g l⁻¹ Poloxamer 188 at 37 °C and 5% $CO_2$. For expression of SBP-tagged Sb1 and for purification of endogenous LRRC8 proteins, cells were grown in BalanCD HEK293 medium (FUJIFILM) supplemented with 1% FBS, 4 mM L-glutamine and 100 U ml⁻¹ penicillin-streptomycin. For electrophysiology, adherent HEK293T and LRRC8[B,D,E–/–] cells were grown in high-glucose DMEM medium (Gibco), supplemented with 10% FBS, 4 mM L-glutamine, 1 mM sodium pyruvate and 100 U ml⁻¹ penicillin-streptomycin.

### Expression and purification of transiently expressed proteins

For transient transfection, a final plasmid DNA concentration of 1.2 µg DNA per ml of cells was used. For co-transfection of LRRC8A and LRRC8C constructs, DNA was added at a 1:1 or 1:3 molar ratio for LRRC8A/C channels in complex with Sb1 and LRRC8A[SAM]/C channels, or at a 1:1.5 molar ratio for LRRC8A/C channels without Sb1. Purified plasmid DNA was mixed with polyethyleneimine (PEI MAX 40 kDa) at a ratio of 1:2.5 (wt/wt) in non-supplemented DMEM medium and incubated at room temperature for 15 min before addition to cells together with 4 mM valproic acid. Transfected cells were incubated for 46–62 h at 37 °C and 5% $CO_2$ in an orbital shaker incubator (Kuhner). Cells were collected, washed with PBS, and the pellets were flash-frozen in liquid nitrogen and stored at −80 °C until further use. Channels used for the structural characterization of LRRC8A/C/Sb1 complexes were expressed in WT HEK293S GnTI⁻ cells. All other transiently transfected proteins were purified from LRRC8⁻/⁻ cells. Protein purification was carried out at 4 °C unless stated differently. For the purification of homomeric LRRC8C or LRRC8A[SAM] channels, cell pellets from 2–3 l of culture were thawed and homogenized by solubilization in 100 ml of lysis buffer (25 mM Tris pH 8.5, 250 mM NaCl, 50 µg ml⁻¹ DNase, 2% digitonin, 10 µM leupeptin, 1 µM pepstatin and 1 µM benzamidine). After 1 h, the lysate was clarified by centrifugation and SBP-tagged proteins were affinity-purified using 5 ml of Streptactin superflow resin (IBA LifeSciences). The resin was washed with 40 column volumes (CVs) of size exclusion chromatography (SEC) buffer 1 (25 mM Tris pH 8.5, 250 mM NaCl and 0.1% digitonin), and bound proteins were eluted with five CVs of SEC buffer 1 supplemented with 15 mM D-desthiobiotin. Tags from eluted proteins were cleaved by incubation with 1.8 mg of human rhinovirus (HRV) 3C protease for 1 h. The protein was concentrated, filtered (0.22-µm filter) and separated on a Superose 6 10/300 GL column (Cytivia), which was equilibrated in SEC buffer 1. Fractions containing the desired protein were pooled and concentrated. Purified proteins were used immediately to prepare samples for cryo-EM.

For crystallization of LRRC8C[LRRD], the protein was expressed in HEK293S GnTI⁻ cells, and purification proceeded by a similar protocol as described in ref. [12], with minor modifications. Cells were solubilized in lysis buffer (10 mM Tris pH 9.4, 200 mM NaCl, 2% n-dodecyl-β-D-maltoside (DDM), 50 µg ml⁻¹ DNase and protease inhibitors (cOmplete EDTA-free, Roche)) and the resin was washed with SEC buffer 2 (10 mM Tris pH 9.4, 200 mM NaCl, 0.1% 3-((3-cholamidopropyl) dimethylammonio)-1-propanesulfonate (CHAPS)) after batch-binding. To release the bound protein from the resin, the slurry was incubated with 0.8 mg HRV 3C protease for 30 min. Eluted protein was concentrated (Amicon, 10 kDa) and separated on a Superdex 75 10/300 column (GE Healthcare) equilibrated with SEC buffer 2. For crystallization, the protein was concentrated to 7.6 mg ml⁻¹ and supplemented with 0.5% CHAPS and 1 mM tri(2-carboxyethyl)phosphine.

For tandem purification of the heteromeric LRRC8A/C or LRRC8A[SAM]/C channels, a similar protocol as described for LRRC8C purification with some modifications was used, as described previously[12]. In all cases, cell pellets from typically 8 l of culture were used. After the clarification of lysates, two consecutive affinity chromatography steps were performed to ensure that the final samples contained both the His-tagged LRRC8C and SBP-tagged LRRC8A or LRRC8A[SAM] constructs. In a first purification step, the lysate was supplemented with 5 mM imidazole and applied to 10 ml of Ni-NTA resin (Agarose Bead Technologies) to pulldown channels containing LRRC8C subunits. The resin was washed with 30 CVs of SEC buffer 1 supplemented with 5 mM imidazole, and the protein was eluted with four CVs of SEC buffer 1 containing 300 mM imidazole. Elution fractions were applied to Streptactin superflow resin (IBA LifeSciences) to remove homomeric LRRC8C channels. All the following steps were performed as described for LRRC8C purification. Final protein samples were either immediately frozen on cryo-EM grids or flash-frozen and used for quantification of both subunits by MS. The typical yield for an A/C tandem purification amounted to 10 µg of protein per litre of transfected cell culture.

### Sybody purification

For the labeling of LRRC8A in samples used for cryo-EM, sybody Sb1 was purified from bacteria as described in ref. [27]. Briefly, the pBXnPH3

plasmid containing the construct coding for Sb1 was transformed into *Escherichia coli* MC1061 and grown in Terrific Broth medium supplemented with ampicillin. Protein expression was induced with arabinose, and bacteria were collected after 19 h. Cells were lysed, and His-tagged Sb1 was purified on Ni-NTA resin (Agarose Bead Technologies). The His-tag was cleaved using HRV 3C protease and the sample was dialyzed overnight to eliminate imidazole. The dialyzed sample was subjected to Ni-NTA resin to remove the cleaved tag before concentration and further purification on a Sepax SRT-10C SEC100 column (Sepax Technologies). Sybody fractions were pooled and concentrated to 5.3 mg ml$^{-1}$, supplemented with glycerol, and flash-frozen until further use. Affinity chromatography, dialysis and concentration steps were performed at room temperature.

### Purification of endogenous LRRC8 channels

Purification of endogenous LRRC8 channels from HEK293T and LRRC8$^{B,D,E-/-}$ cells was performed by one of two strategies. In one protocol, the respective cells were transfected with DNA coding for the SBP-tagged sybody Sb1, and endogenous LRRC8 channels were pulled down during sybody purification. Channels purified in this way were only used for subunit quantification and not for structural characterization. In the alternate approach, endogenous LRRC8 channels were isolated from non-transfected cell pellets (grown to a density of up to five million cells per ml) using purified SBP-tagged Sb1 bound to Streptactin superflow resin as the affinity matrix. For affinity purification from HEK293T and LRRC8$^{B,D,E-/-}$ cells, the pcDXc3MS plasmids containing the construct for Sb1 were transfected into HEK293S GnTI$^-$ cells grown in suspension culture as described above, and cells were collected at 48–62 h post transfection. For each round of protein preparation, typically 10–20 g of non-transfected cells or pellets from 1–3 l of transfected cells were used. For purification, cell pellets were disrupted in lysis buffer (36 mM Tris pH 8.5, 150 mM NaCl, 2% glyco-diosgenin (GDN), 50 µg ml$^{-1}$ DNase, 10 µM leupeptin, 1 µM pepstatin and 1 µM benzamidine) for 1–2.5 h before clarifying the lysate by centrifugation. Lysates from cells transfected with Sb1 were directly loaded onto 100–300 µl of Streptactin superflow resin (IBA LifeSciences) to purify the SBP-tagged sybodies. For affinity purification using resin containing immobilized purified sybody, 100 µl of Streptactin superflow resin was loaded with 400 µg of SBP-tagged Sb1 and incubated for 30 min before loading the clarified lysate. In both approaches, the resin was washed with SEC buffer 3 (36 mM Tris pH 8.5, 150 mM NaCl and 0.0105% GDN), and the protein was eluted with the same buffer supplemented with 15 mM D-desthiobiotin. Fractions containing the desired protein were concentrated and injected onto a Superose 6 10/300 GL column (Cytivia) pre-equilibrated in SEC buffer 3. Peak fractions at the elution volume of LRRC8 channels were collected, and the LRRC8A subunit was detected by western blot using an anti-LRRC8A antibody (Sigma, SAB1412855, 1:1,000 dilution). The concentrated peak fractions were either used directly for the preparation of cryo-EM grids or flash-frozen and stored at −80 °C for a later quantification by MS. The typical yield was 0.35 µg of protein per gram of LRRC8$^{B,D,E-/-}$ cell pellet.

### Sample preparation for liquid chromatography with tandem mass spectrometry

Samples for the quantification of LRRC8A and LRRC8C subunits were collected either before or after the final SEC step from the protein preparations described above. For endogenous channels, samples were measured from either two or six independent protein preparations purified from WT or LRRC8$^{B,C,E-/-}$ cells, respectively. For transiently expressed LRRC8A/C channels, three independent preparations from cells transfected at a 1:1 DNA ratio were characterized, one obtained from HEK293S GnTI$^-$ cells and two from LRRC8$^{-/-}$ cells. For 1:3 transfected channels and LRRC8$^{SAM}$/C channels (both expressed in LRRC8$^{-/-}$ cells), the sample obtained from a single purification was analyzed.

Samples for LC-MS/MS analysis were processed using a commercial iST Kit (PreOmics). For each sample, 0.5–1 µg of protein was mixed with 'Lyse' buffer, boiled at 95 °C for 10 min, transferred to the cartridge and digested by adding 50 µl of the 'Digest' solution. After 2 h of incubation at 37 °C, the digestion was stopped with 100 µl of 'Stop' solution. The solutions in the cartridge were removed by centrifugation at 3,800$g$, and the peptides were retained by the iST-filter. Finally, the peptides were washed, eluted, dried and re-solubilized in 20 µl of MS sample buffer (3% acetonitrile, 0.1% formic acid). Samples used for absolute quantification were spiked with heavy labeled peptides (25–500 fmol depending on the sample) before digestion.

Peptide sequences for absolute quantification included the sequences IEAPALAFLR (residues 483–492) and YIVIDGLR (residues 534–541) of LRRC8A and NSLSVLSPK (residues 739–747) and NSLSVLSPK (residues 739–747) of LRRC8C. These were selected based on the criteria of peptide length (7–25 amino acids) and the absence of methionines, cysteines and ragged ends (KR/RR). In addition, the selected tryptic peptides covered shared sequences in the mouse and human genome and were found to exhibit a linear response in the dynamic range of the detector. Absolute quantified, stable-isotope labeled peptides (SIL) were synthesized as SpikeTides TQL at >95% purity by JPT Peptide Technology, as determined by HPLC, MS and amino acid analysis. C-terminal lysines or arginines were incorporated as heavily labeled amino acids (Arg:U-13C6; U-15N4; Lys: U-13C6; U-15N2).

### Liquid chromatography with tandem mass spectrometry analysis

The MS analysis was performed on an Orbitrap Exploris 480 mass spectrometer (Thermo Fisher Scientific) equipped with a Nanospray Flex Ion Source (Thermo Fisher Scientific) and coupled to an M-Class UPLC system (Waters). Solvent composition at the two channels was 0.1% formic acid for channel A and 0.1% formic acid, 99.9% acetonitrile for channel B. The column temperature was 50 °C. For each sample, 1 µl of peptides was loaded on a commercial nanoEase MZ Symmetry C18 trap column (100 Å, 5 µm, 180 µm × 20 mm, Waters) followed by a nanoEase MZ C18 HSS T3 column (100 Å, 1.8 µm, 75 µm × 250 mm, Waters). The peptides were eluted at a flow rate of 300 nl min$^{-1}$. After a 3-min initial hold at 5% B, a gradient from 5 to 35% B was applied over 60 min. The column was cleaned after the run by increasing to 95% B and holding at 95% B for 10 min before re-establishing the loading condition for another 10 min.

For absolute quantification, the mass spectrometer was operated in parallel reaction monitoring mode with a scheduled (5-min windows) inclusion list using Xcalibur 4.5 (Tune version 4.0), with spray voltage set to 2.3 kV, funnel RF level of 40%, heated capillary temperature of 275 °C, and 'advanced peak determination' on. Full-scan MS spectra (350–1,500 $m/z$) were acquired at a resolution of 120,000 at 200 $m/z$ after accumulation to a target value of 3,000,000 or for a maximum injection time of 50 ms. Precursors of heavy and light peptides were selected as stated in Extended Data Fig. 1d, isolated using a quadrupole mass filter with 1-$m/z$ isolation window and fragmented by higher-energy collisional dissociation (HCD) using a normalized collision energy of 30%. HCD spectra were acquired at a resolution of 30,000, and the maximum injection time was set to Auto. The automatic gain control was set to 100,000 ions. The samples were acquired using an internal lock mass calibration on $m/z$ 371.1012 and 445.1200.

Data-dependent acquisition for the identification of endogenous LRRC8 peptides in isolated complexes was performed with full-scan MS spectra (350–1,200 $m/z$) acquisition at a resolution of 120,000 after accumulation to a target value of 3,000,000, followed by HCD fragmentation for a cycle time of 3 s. Ions were isolated with a 1.2-$m/z$ isolation window and fragmented by HCD using a normalized collision energy of 30%. HCD spectra were acquired at a resolution of 30,000 and a maximum injection time of 119 ms. The automatic gain control

was set to 100,000 ions. Precursor masses previously selected for MS/MS measurement were excluded from further selection for 20 s, and the exclusion window tolerance was set at 10 ppm. The MS proteomics data were handled using the local laboratory information management system[48].

## Protein and peptide identification and quantification

For the generation of spectral libraries from SIL peptides, acquired raw MS data were converted into Mascot Generic Format files (.mgf) using Proteome Discoverer 2.1, and the proteins were identified using the Mascot search engine (Matrix Science, version 2.7). Spectra were searched against a reviewed UniProt proteome database (taxonomy 9606, version from 9 July 2019), concatenated to its reversed decoyed fasta database. Enzyme specificity was set to trypsin and modification to 13C(6)15N(2) for lysine and 13C(6)15N(4) for arginine. A fragment ion mass tolerance of 0.02 Da and a parent ion tolerance of 10 ppm were set. Scheduled parallel reaction monitoring runs of spiked samples were imported into Skyline. Identity assignments were evaluated by determining spectra similarity between endogenous and SIL peptides via dot product. Endogenous peptide quantification was carried out by one-point calibration using the ratio of the endogenous and SIL peptides and is given in units of fmol on column[49]. For each peptide, at least four transitions were used for quantification. The protein ratio estimation was based on the median of all combinations of pairwise peptide ratios between LRRC8A and LRRC8C, similar to the protein ratio algorithm employed by MaxQuant[50]. For data-dependent acquisition data, raw MS data were converted into .mgf format using Proteome Discoverer 2.1, and the proteins were identified using the Mascot search engine (Matrix Science, version 2.7.0.1). Spectra were searched against the UniProt *Homo sapiens* reference proteome (taxonomy 9606, canonical version from 9 July 2019), concatenated to its reversed decoyed fasta database and common protein contaminants. Carbamidomethylation of cysteine was set as a fixed modification, and methionine oxidation and N-terminal protein acetylation were set as variables. Enzyme specificity was set to trypsin/P, allowing a maximum of two missed cleavages. Scaffold (Proteome Software Inc., version 5.10) was used to validate MS/MS-based peptide and protein identifications. Peptide identifications were accepted if they achieved a false discovery rate (FDR) of less than 0.1% by the Scaffold Local FDR algorithm. Protein identifications were accepted if they achieved an FDR of less than 1.0% and contained at least two identified peptides. Peptide and spectral counts are provided in Extended Data Fig. 1a.

## X-ray structure determination of the LRR domain of LRRC8C

The C-terminal domain construct LRRC8C[LRRD], containing three additional residues at the N terminus (Gly-Pro-Ser) and an additional alanine at the C terminus, was crystallized by vapor diffusion in sitting drops at 4 °C. Drops were prepared by mixing 0.1 µl of protein solution with 0.1 µl of precipitant solution containing 0.2 M magnesium chloride and 20% PEG3350 (NeXtal PACT Suite, Qiagen). Crystals were collected after two weeks, cryo-protected in crystallization solution containing an additional 30% of ethylene glycol, and flash-frozen in liquid nitrogen. X-ray diffraction data were collected on the X10SA beamline at the Swiss Light Source of the Paul-Scherrer Institute on a Pilatus 6M detector. Data were collected from a single crystal at a wavelength of 1 Å and processed with XDS[51]. The crystals are of space group $P2_1$ and contain four copies of the molecule in their asymmetric unit (Extended Data Fig. 2a,b). Initial phases were obtained by molecular replacement with Phaser[52], implemented in the Phenix suite[53] using the structure of the LRRD of LRRC8A (PDB 6FNW) as the search model. The structure was built in Coot[54] and improved by iterative cycles of manual corrections and refinement with Phenix. $R_{free}$ was calculated based on 5% of reflections excluded from refinement. The final model consisting of 1,618 residues is well-refined, with $R_{work}$ and $R_{free}$ values of 24% and 29%, respectively (Extended Data Fig. 2a).

## Cryo-electron microscopy sample preparation and data collection

Cryo-EM grids were frozen immediately after purification. For grids of LRRC8A/C from transiently expressed proteins, samples were concentrated to a final concentration of 2–5 mg ml$^{-1}$. Endogenous LRRC8A/C channels were concentrated to a total protein concentration of 0.075–0.2 mg ml$^{-1}$. For the analysis of these channels, purified, tag-free Sb1 was added to the purified complexes at a 1:1.5 molar excess (based on all LRRC8 subunits) directly before grid freezing. Homomeric LRRC8C was concentrated to 5 mg ml$^{-1}$. Aliquots were either frozen directly or after addition of Sb1 at a 1:1.5 molar excess (per LRRC8C subunit) as a negative control. For LRRC8A/C channels not containing Sb1, a different purification approach was chosen. The low-affinity binder Sb3 (ref. [27]) was added after affinity purification of the heteromeric channel as an attempt to introduce an alternate label to the A subunits. The sample was then concentrated to 2 mg ml$^{-1}$ and subjected to SEC. After SEC and the following concentration of peak fractions, Sb3 was no longer present in the sample (as confirmed by SDS–PAGE and the absence of sybody density in the cryo-EM structure). For grids of LRRC8A$^{SAM}$/C, the sample was concentrated to 4 mg ml$^{-1}$. For vitrification of transiently expressed proteins, 2–2.5 µl of protein solution was applied to glow-discharged holey carbon grids (Quantifoil R1.2/1.3 Au 200 mesh). The endogenous samples were applied onto grids containing an additional thin layer of continuous carbon support (Quantifoil R1.2/1.3 Cu 200 mesh + 2 nm C) and incubated for 30 s before blotting and freezing. Excess sample was removed by blotting grids for 3–5 s with 0 blotting force in a controlled environment (4 °C, 100% humidity). Grids were flash-frozen in a mixture of liquid ethane/propane using a Vitrobot Mark IV system (Thermo Fisher Scientific) and stored in liquid nitrogen until further use. Samples were imaged on a 300-kV Titan Krios G3i set-up (Thermo Fisher Scientific) with a 100-µm objective aperture. All data were collected using a post-column BioQuantum energy filter (Gatan) with a 20-eV slit and a K3 direct detector (Gatan) operating in super-resolution mode. Dose-fractionated micrographs were recorded with a defocus range of −1.0 to −2.4 µm in automated mode using EPU 2.9 (Thermo Fisher Scientific). Data were recorded at a nominal magnification of ×130,000 corresponding to a pixel size of 0.651 Å per pixel (0.3255 Å per pixel in super-resolution) with a total exposure time of either 1 s (36 individual frames) with a dose of -1.85 e$^-$ per Å$^2$ per frame or 1.26 s (47 individual frames) and a dose of -1.26 e$^-$ per Å$^2$ per frame. The total electron dose on the specimen level for all datasets was ~67 e$^-$ Å$^{-2}$ and 59 e$^-$ Å$^{-2}$, respectively. Several of the described large datasets consist of a few smaller datasets collected on multiple occasions. Briefly, the LRRC8C structure was obtained from three datasets containing 13,333, 5,942 and 14,301 micrographs. The sample used for the collection of the latter two datasets contained Sb1 as control, which, as expected, was not bound and did not influence the conformational properties of the sample. The LRRC8A/C$^{1:1}$/Sb1 structure was obtained from two datasets consisting of 10,800 and 15,642 micrographs, respectively. The LRRC8A/C$^{1:3}$/Sb1 structure was obtained after merging two datasets containing 19,911 and 13,761 micrographs each. The endogenous LRRC8A/C/Sb1 structure was obtained from three datasets containing 6,681, 22,254 and 19,053 micrographs. The LRRC8A$^{SAM}$/C was determined from a single dataset of 24,560 micrographs. LRRC8A/C with the unlabeled A subunit was determined from one dataset containing 14,160 micrographs. The LRRC8A$^{SAM}$ dataset was determined from 1,677 micrographs collected on a 300-kV Tecnai G2 Polara microscope (FEI) with a 100-µm objective aperture using a post-column quantum energy filter (Gatan) with a 20-eV slit and a K2 Summit direct detector (Gatan) operating in counting mode.

## Cryo-electron microscopy image processing

All data processing was performed in RELION 3.1.2 and RELION 4.0-beta[55,56] by a general procedure similar to that described in the

following. Detailed information and processing steps relevant to a specific dataset are included in Extended Data Figs. 3, 4 and 7–9. In all datasets, acquired super-resolution images were gain-corrected and down-sampled twice using Fourier cropping, resulting in a pixel size of 0.651 Å. All frames were used for beam-induced motion correction with a dose-weighting scheme using RELION's own implementation of the MotionCor2 program[57]. CTF parameters were estimated using CTFFIND4.1 (ref. [58]). Micrographs showing a large drift, high defocus or poor CTF estimates were removed. Particles were autopicked using templates generated from a previously reported dataset of full-length LRRC8A/Sb1 (ref. [27]). Particles were extracted with a box size of 672 pixels and compressed four times (168-pixel box size, 2.604 Å per pixel) for initial processing. Extracted particles were subjected to two rounds of reference-free 2D classification. As datasets of LRRC8C, LRRC8A/C$^{1:1}$/Sb1, LRRC8A/C$^{1:3}$/Sb1 and endogenous LRRC8A/C/Sb1 consist of combined smaller datasets, cleaned-up particles from the respective individual datasets were merged before being subjected to two rounds of 3D classification. To preserve the unique structural features of the LRRC8A and C subunits, 3D classification and 3D auto-refinement were always carried out with $C1$ symmetry applied unless stated otherwise. For the first 3D classification, a previously determined map of LRRC8A/Sb1 (ref. [27]) was used as a reference after low-pass filtering to 60 Å. In further processing steps, the respective best maps at each stage were used as references after low-pass filtering to 40 Å. Particles subjected to 3D auto-refinement were masked with soft masks encompassing only protein density. In datasets of LRRC8C and LRRC8A/C$^{1:1}$/Sb1, when the reported resolution reached the Nyquist limit, selected particles were re-extracted with twofold binning (336-pixel box size, 1.302 Å per pixel) and subjected to iterative 3D auto-refinement, per-particle motion correction[59] and per-particle CTF correction[55]. To improve the resolution of the LRRC8C channel, $C7$ symmetry was applied during 3D refinement of the full-length protein, as well as the focus-refinement of the TM domain. In the LRRC8A/C$^{1:1}$/Sb1 dataset, polished particles were subjected to further iterative 3D classification in $C1$ without alignment, followed by refinement to separate assemblies of A:C ratio other than 2:1. Despite two other low-resolution classes emerging, the predominant class displayed a 2:1 arrangement. Masked local refinement maintaining $C1$ symmetry of the TM domain, the ESD containing the selectivity filter and a pair of tightly interacting LRRDs from A subunits with bound Sb1 increased the resolution of these regions compared to the resolution of the full-length channel. The same approach of masked 3D classification without particle alignment was applied for the LRRC8A$^{SAM}$/C dataset, but it did not discriminate between the alternate arrangements of the A and C subunits. The resolution of all generated maps was estimated using a soft solvent mask and based on the gold-standard Fourier shell correlation (FSC) 0.143 criterion[60–62]. The cryo-EM densities, except for the endogenous LRRC8A/C$^{endog}$/Sb1, were also sharpened using isotropic b-factors.

**Cryo-electron microscopy model building and refinement**
All models of LRRC8 channels and their sybody complexes were built into cryo-EM density with Coot[54] and improved by real-space refinement in PHENIX[63]. For the LRRC8C homomer, a homology model of the PD of LRRC8C (generated by SWISS-MODEL[64]) was rebuilt into 4.1-Å density of the masked PD of LRRC8C and improved by alternating cycles of refinement and manual corrections. Subsequently, the refined X-ray structure of LRRC8C$^{LRRD}$ was inserted into the low-resolution density envelope of the respective region of the entire channel at 4.6 Å, and its position was initially improved by rigid body refinement, followed by a few cycles of all-atom refinement in PHENIX applying strong NCS constraints. Model building into the map of the LRRC8A/C$^{1:1}$/Sb1 complex at 3.8 Å was initiated by placing the four LRRC8A subunits with bound Sb1 into the density using the structure of the homomeric LRRC8A/Sb1 complex (PDB 7P5V) as template. The coordinates of the

ESD and the TM of the two LRRC8C subunits were obtained from the refined LRRC8C PD structure, whereas the CSDs, and the LRRDs of the two C-chains, are mobile and thus not defined in the density. The model was improved by alternating rebuilding and refinement cycles without imposing NCS symmetry constraints. Parts of the model were separately refined and improved in masked maps obtained for the PD, the ESD and the LRRD/Sb1 dimer. For refinement of the LRRC8A/C structure into density of the complex obtained in the absence of Sb1 at 6.6 Å, the refined channel component of the LRRC8A/C/Sb1 complex was fitted into the density. The LRRC8C subunits in the C1 and C2 positions were introduced from the LRRC8C homomer where the LRRD of the subunit in the C2 position was deleted. The altered orientations of the LRRDs were initially fitted manually, then improved by rigid body refinement in PHENIX, treating the PDs and LRRDs as separate units. The full-length subunits (except for C2, where the LRRD was not defined) were improved in a final step of refinement. Pore radii were determined with HOLE[41]. Figures were prepared with DINO (http://www.dino3d.org), Chimera[65] and ChimeraX[66]. Surfaces were generated with MSMS[67].

**Electrophysiology**
For electrophysiology, HEK293T, HEK293 LRRC8$^{B,D,E-/-}$ and HEK293 LRRC8$^{-/-}$ cells were seeded into Petri dishes at 3% confluency on the day before the measurements. For recordings from overexpressed protein, HEK293 LRRC8$^{-/-}$ were transfected with 1.2 µg LRRC8A or LRRC8A$^{SAM}$, 1.2 µg LRRC8C and 1.6 µg Venus-only plasmids per 6-cm dish, 4–5 h after seeding and 14 h before analysis using Lipofectamine 2000 (Invitrogen). All measurements were performed at 20 °C. Patch pipettes were pulled from borosilicate glass capillaries (inner diameter of 0.86 mm and outer diameter of 1.5 mm) with a micropipette puller (Sutter) and fire-polished with a Microforge (Narishige). The typical pipette resistance was 2–7.5 MΩ when filled with pipette solution composed of 10 mM HEPES-NMDG pH 7.4, 150 mM NMDG-Cl, 1 mM EGTA and 2 mM Na$_2$ATP (266 mmol kg$^{-1}$). Seals with a resistance of 4 GΩ or higher were used to establish the whole-cell configuration. Data were recorded with an Axopatch 200B amplifier and digitized with Digidata 1440 (Molecular Devices). Analog signals were digitized at 10–20 kHz and filtered at 5 kHz using the in-built four-pole Bessel filter. Data acquisition was performed using the Clampex 10.6 software (Molecular Devices). Cells were locally perfused using a gravity-fed system. After break-in into the cell and establishment of the whole-cell configuration, cells were perfused with isotonic buffer (10 mM HEPES-NMDG pH 7.4, 95 mM NaCl, 1.8 mM CaCl$_2$, 0.7 mM MgCl$_2$ and 100 mM mannitol, 298 mmol kg$^{-1}$). After 1 min, cell swelling was initiated by switching the perfusion buffer to hypotonic buffer (10 mM HEPES-NMDG pH 7.4, 95 mM NaCl, 1.8 mM CaCl$_2$ and 0.7 mM MgCl$_2$, 194 mmol kg$^{-1}$). Approximately 85% of WT cells and 35% of LRRC8$^{B,D,E-/-}$ cells showed current response upon swelling. HEK293 LRRC8$^{-/-}$ cells showed a current response in 90% and 70% of the patched cells when transfected with LRRC8A and LRRC8C or LRRC8A$^{SAM}$ and LRRC8C, respectively. Currents were monitored in 2-s intervals for 6–7 min using a ramp protocol (15 ms at 0 mV, 100 ms at −100 mV, a 500-ms linear ramp from −100 mV to 100 mV, 100 ms at 100 mV, 200 ms at −80 mV, 1,085 ms at 0 mV). The values at 100 mV, 10 ms after the ramp, are displayed in the activation curves. Current–voltage relationships ($I$–$V$) were obtained from a voltage-jump step protocol (from −100 to 120 mV in 20-mV steps). Current rundown was corrected using a pre-pulse recorded at −80 mV preceding each voltage ramp. After the voltage-jump step protocol, cells were perfused with hypotonic buffer for an additional 20–30 s before switching to isotonic solution, initiating cell shrinkage. Inactivation of currents was monitored with the same ramp protocol as described above. For measurements in hypotonic conditions, only one cell was used per dish. Data were analyzed using Clampfit 10.6 (Molecular Devices) and Excel (Microsoft).

## Statistics and reproducibility

Electrophysiology data were repeated multiple times from different cells with comparable results. Conclusions of experiments were not changed upon inclusion of further data. In all cases, leaky patches were discarded.

## Reporting summary

Further information on research design is available in the Nature Portfolio Reporting Summary linked to this Article.

## Data availability

The 3D cryo-EM density maps have been deposited in the Electron Microscopy Data Bank under accession nos. EMD-15835 (LRRC8C), EMD-15836 (LRRC8A/C$^{1:1}$/Sb1), EMD-15837 (LRRC8A/C), EMD-15838 (LRRC8A/C$^{1:3}$/Sb1), EMD-15839 (LRRC8A/C$^{endog}$/Sb1), EMD-15840 (LRRC8A$^{SAM}$) and EMD-15841 (LRRC8A$^{SAM}$/C). The deposition includes maps of full-length proteins, both corresponding half-maps, the mask used for final FSC calculations, as well as relevant higher-resolution maps obtained after local refinement. Coordinates have been deposited in the Protein Data Bank under accession numbers 8B40 (LRRC8C), 8B41 (LRRC8A/C$^{1:1}$/Sb1) and 8B42 (LRRC8A/C). Coordinates and structure factors of the X-ray structure of the LRRD of LRRC8C have been deposited in the PDB under accession no. 8BEN. The MS proteomics data have been deposited to the ProteomeXchange Consortium via the PRIDE (http://www.ebi.ac.uk/pride) partner repository with the dataset identifier PXD035350. Source data are provided with this paper.

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

## Acknowledgements

This research was supported by a grant from the Swiss National Science Foundation (no. 310030_204373 to R.D.). We thank the Center for Microscopy and Image Analysis (ZMB) of the University of Zurich for support and access to electron microscopes, and T. J. Jentsch from FMP/MDC, Berlin for providing the LRRC8$^{−/−}$ and LRRC8$^{B,D,E−/−}$ HEK cell lines. We also thank L. Kunz for her support and scientific input for MS experiments. All members of the Dutzler laboratory are acknowledged for their help at various stages of the project. X-ray diffraction data were collected on the X10SA beamline at the Swiss Light Source of the Paul-Scherrer Institute.

## Author contributions

S.R. expressed and purified proteins and prepared MS samples. A.D. carried out MS analysis. D.D. prepared constructs and determined the X-ray structure of the LRRD of LRRC8C and the structure of the LRRC8A$^{SAM}$ construct. S.R. and M.S. prepared samples for cryo-EM and collected cryo-EM data, and proceeded with structure determination and refinement. S.R. recorded and analyzed electrophysiology data. S.R., M.S., D.D., A.D. and R.D. jointly planned the experiments, analyzed the data and wrote the manuscript.

## Competing interests

The authors declare no competing interests.

## Additional information

**Extended data** is available for this paper at

**Supplementary information** The online version
contains supplementary material available at

**Correspondence and requests for materials** should be addressed to
Marta Sawicka or Raimund Dutzler.

**Peer review information** *Nature Structural & Molecular Biology*
thanks Zhaozhu Qiu and the other, anonymous, reviewer(s) for their
contribution to the peer review of this work. Primary Handling Editor:
Carolina Perdigoto, in collaboration with the *Nature Structural &
Molecular Biology* team. Peer reviewer reports are available.

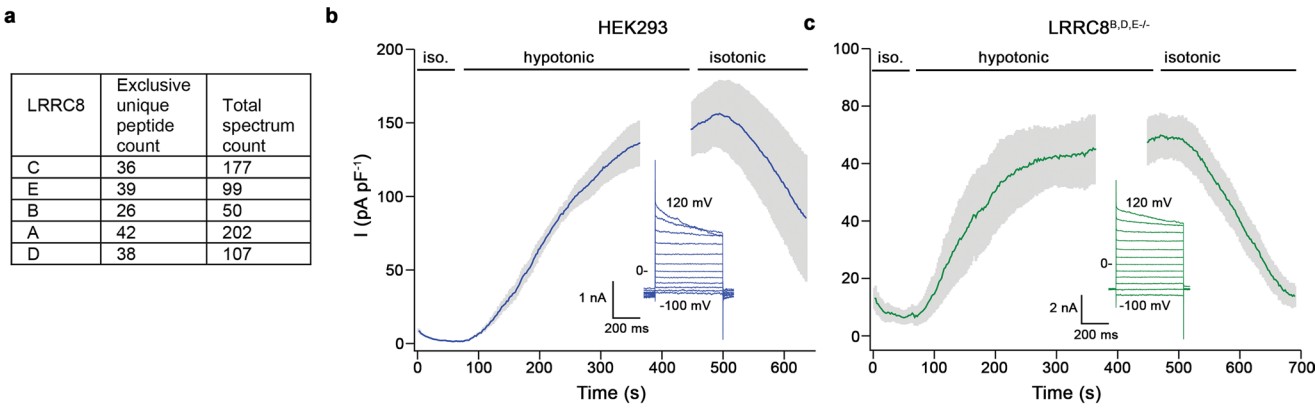

**a**

| LRRC8 | Exclusive unique peptide count | Total spectrum count |
|---|---|---|
| C | 36 | 177 |
| E | 39 | 99 |
| B | 26 | 50 |
| A | 42 | 202 |
| D | 38 | 107 |

**d**

| Protein | Peptide sequence | Start position Mouse | End position Mouse | Precursor m/z | Precursor charge | Fragment ion | Fragment ion m/z |
|---|---|---|---|---|---|---|---|
| LRRC8A | IEAPALAFLR | 483 | 492 | 550.8268 | 2 | y9 | 987.5622 |
| | | | | | | y8 | 858.5196 |
| | | | | | | y7 | 787.4825 |
| | | | | | | y6 | 690.4297 |
| | | | | | | y5 | 619.3926 |
| | IEAPALAFLR* | | | 555.8309 | | y9* | 997.5705 |
| | | | | | | y8* | 868.5279 |
| | | | | | | y7* | 797.4908 |
| | | | | | | y6* | 700.4380 |
| | | | | | | y5* | 629.4009 |
| | YIVIDGLR | 534 | 541 | 474.7793 | 2 | y7 | 785.4880 |
| | | | | | | y6 | 672.4039 |
| | | | | | | y5 | 573.3355 |
| | | | | | | y4 | 460.2514 |
| | YIVIDGLR* | | | 479.7834 | | y7* | 795.4962 |
| | | | | | | y6* | 682.4122 |
| | | | | | | y5* | 583.3438 |
| | | | | | | y4* | 470.2597 |
| LRRC8C | YLDLSYNDIR | 687 | 696 | 636.3170 | 2 | y9 | 1108.5633 |
| | | | | | | y8 | 995.4793 |
| | | | | | | y7 | 880.4523 |
| | | | | | | y6 | 767.3682 |
| | | | | | | y5 | 680.3362 |
| | | | | | | y4 | 517.2729 |
| | YLDLSYNDIR* | | | 641.3211 | | y9* | 1118.5716 |
| | | | | | | y8* | 1005.4875 |
| | | | | | | y7* | 890.4606 |
| | | | | | | y6* | 777.3765 |
| | | | | | | y5* | 690.3445 |
| | | | | | | y4* | 527.2812 |
| | NSLSVLSPK* | 739 | 747 | 472.7742 | 2 | y8 | 830.4982 |
| | | | | | | y7 | 743.4662 |
| | | | | | | y6 | 630.3821 |
| | | | | | | y5 | 543.3501 |
| | | | | | | y4 | 444.2817 |
| | NSLSVLSPK* | | | 476.7813 | | y8* | 838.5124 |
| | | | | | | y7* | 751.4804 |
| | | | | | | y6* | 638.3963 |
| | | | | | | y5* | 551.3643 |
| | | | | | | y4* | 452.2959 |

**e**

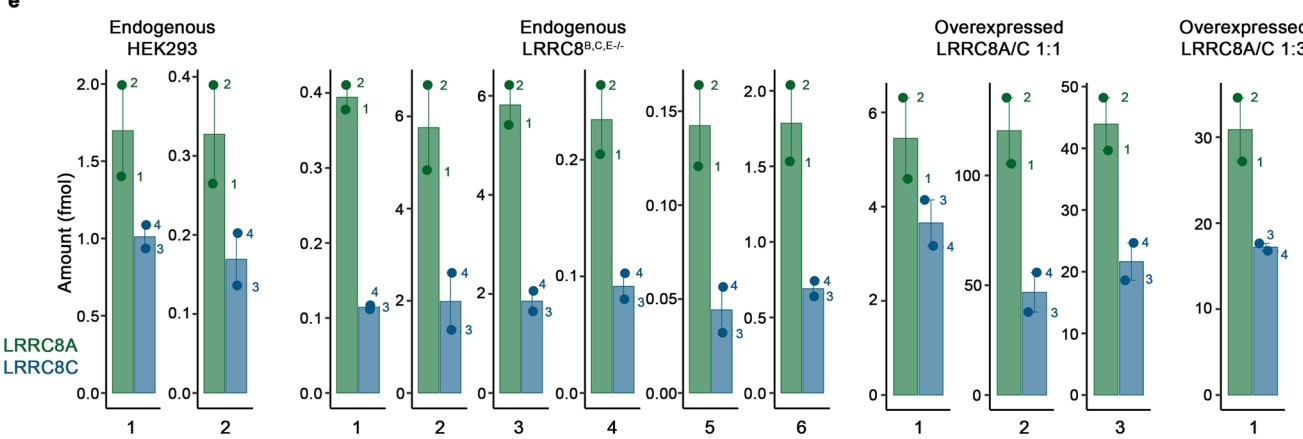

**Extended Data Fig. 1 | See next page for caption.**

**Extended Data Fig. 1 | Mass spectrometry and electrophysiology. a**, Identification of all five subunits in isolated endogenous LRRC8 complexes from HEK293T cells by shotgun LC-MS/MS. **b**, **c**, VRAC currents recorded in the whole-cell configuration in response to a change of the extracellular environment to hypotonic conditions. Values show average of three biological replicates recorded at 100 mV in, **b**, HEK293T and, **c**, LRRC8[B,D,E,-/-] cells. Errors are s.e.m. The changes in extracellular osmolarity are indicated (top). Insets show traces of a representative cell recorded at different voltages at the plateau of activation. **d**, LRRC8A/C precursor and product ion masses used for absolute quantification. Asterisk denotes heavy labeled residues. **e**, Determination of absolute peptide levels in isolated LRRC8 complexes using LC-MS/MS. Peptide amounts on column (fmol) were calculated using the ratio of endogenous and SIL (stable-isotope labeled) peptides: IEAPALAFLR (1) and YIVIDGLR (2) from LRRC8A and NSLSVLSPK (3) and YLDLSYNDIR (4) from LRRC8C. Graphs show individual biological replicates with bars representing the average of the two peptides belonging to the same subunit.

**a**

**Data collection and refinement statistics**

|  | LRRC8C$^{LRRD}$ |
|---|---|
| **Data collection** | |
| Space group | P2$_1$ |
| Cell dimensions | |
| $a, b, c$ (Å) | 73.4, 204.6, 111.9 |
| $\alpha, \beta, \gamma$ (°) | 90, 94.7, 90 |
| Resolution (Å) | 49-3.1 (3.2-3.1) * |
| $R_{sym}$ or $R_{merge}$ | 35.0 (91.5) |
| $I / \sigma I$ | 5.5 (2.2) |
| Completeness (%) | 98.3 (98.9) |
| Redundancy | 7.2 (7.4) |
| | |
| **Refinement** | |
| Resolution (Å) | 49.2-3.1 |
| No. reflections | 30437 (3056) |
| $R_{work} / R_{free}$ | 24/29 |
| No. atoms | 13,038 |
| Protein | 13,038 |
| Ligand/ion | 0 |
| Water | 0 |
| $B$-factors | 26.3 |
| Protein | 26.3 |
| Ligand/ion | - |
| Water | - |
| R.m.s. deviations | |
| Bond lengths (Å) | 0.007 |
| Bond angles (°) | 0.84 |

*Values in parentheses are for highest-resolution shell.

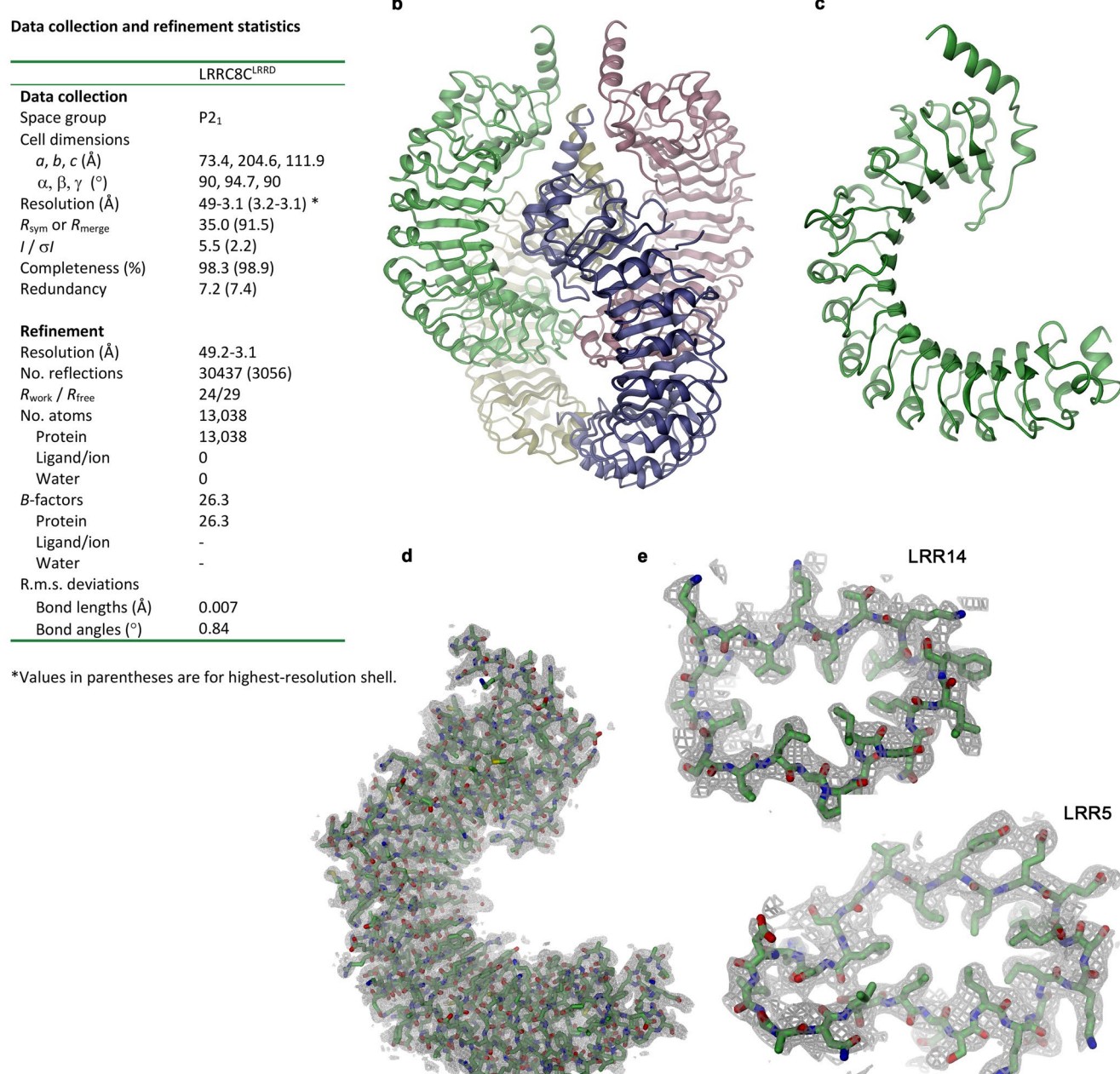

**Extended Data Fig. 2 | X-ray structure of the LRRC8C domain. a**, X-ray data collection and refinement statistics of the LRRC8C$^{LRRD}$ construct. **b**, Structure of the asymmetric unit of LRRC8C$^{LRRD}$ crystals containing four copies of the protein. **c**, Structure of a single LRRC8C$^{LRRD}$ chain. **b**, **c**, Proteins are displayed as ribbon. **d**, Structure of the entire LRRC8C$^{LRRD}$ construct and **e**, of the leucine rich repeats 14 (left) and 5 (right). **d**, **e**, Proteins are shown as sticks with electron density at 3.1 Å (contoured at 0.9 σ) superimposed as gray mesh.

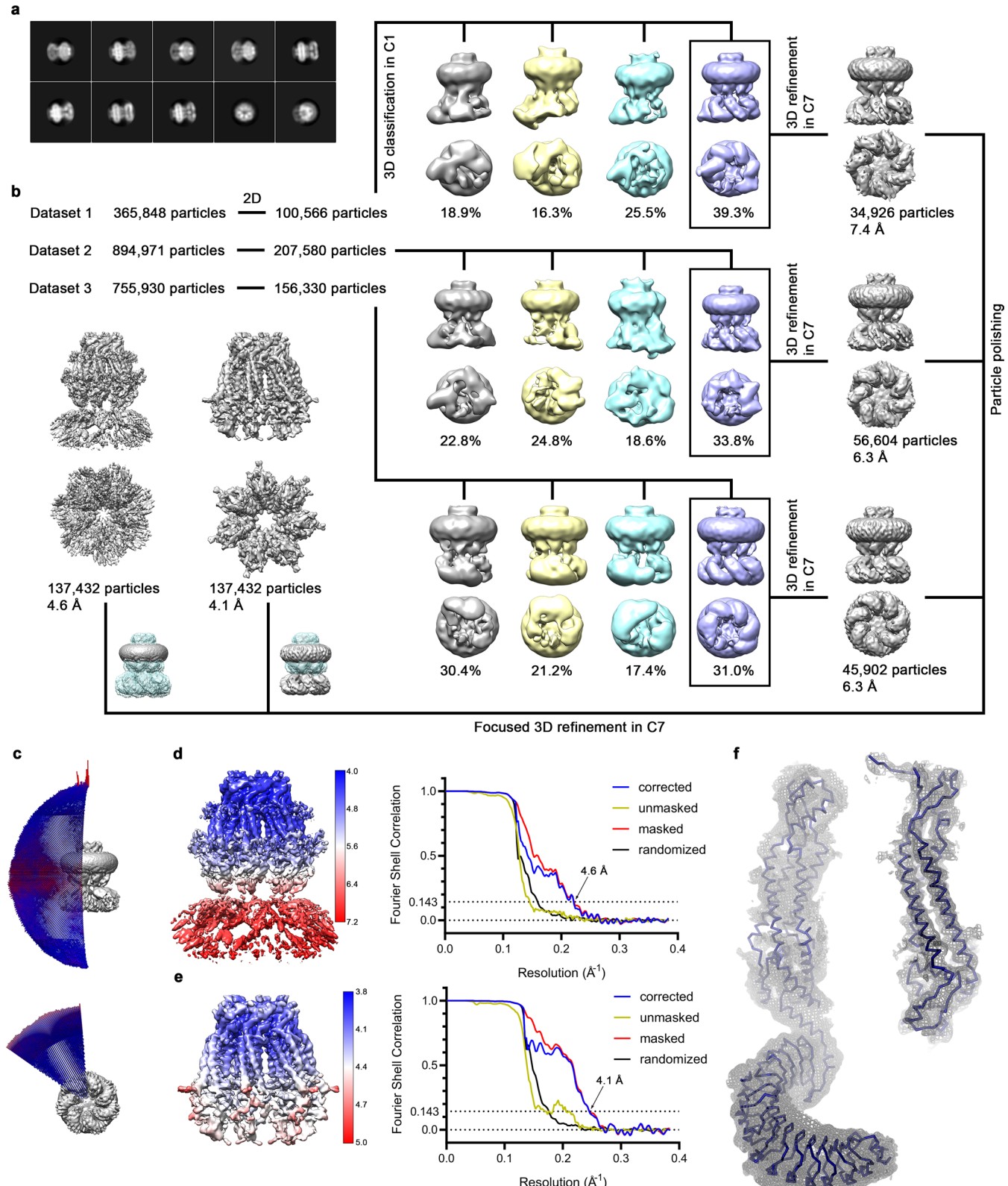

**Extended Data Fig. 3 | See next page for caption.**

**Extended Data Fig. 3 | Cryo-EM structure of LRRC8C. a**, Representative 2D class averages of the LRRC8C channel. **b**, Data processing workflow. Particles from three datasets were cleaned-up during 2D classification and selected class averages showing high resolution features were used in the subsequent 3D classification with no symmetry applied. In each dataset, one out of four classes showed a symmetric pore domain and seven, albeit flexible, LRRDs. The distribution of particles (%) is indicated. Particles assigned to the boxed classes were used in independent refinements with C7 symmetry applied. As all three reconstructions show the same features, all particles belonging to these refined models were pooled and subjected to per-particle motion correction. Polished particles were subsequently used as input for C7-symmetrized focused refinement of the full-length channel and TM region. Insets show the masked

regions during refinement. **c**, Angular distribution plot of all particles included in the final reconstruction of the full-length complex. The length and color of cylinders correspond to the number of particles with respective Euler angles. **d, e**, Final 3D reconstruction colored according to local resolution (left) and FSC plot (right) of the final refined unmasked (yellow), masked (red), phase-randomized (black) and corrected for mask convolution effects (blue) cryo-EM density map of the LRRC8C channel. The resolution at which the FSC curve drops below the 0.143 threshold is indicated. **d**, Full-length complex and **e**, masked PD. **f**, Cα representation of a single subunit of LRRC8C with cryo-EM density of the entire protein at 4.6 Å (contoured at 5 σ, left) and of its PD at 4.1 Å (contoured at 10 σ, right) shown superimposed as gray mesh.

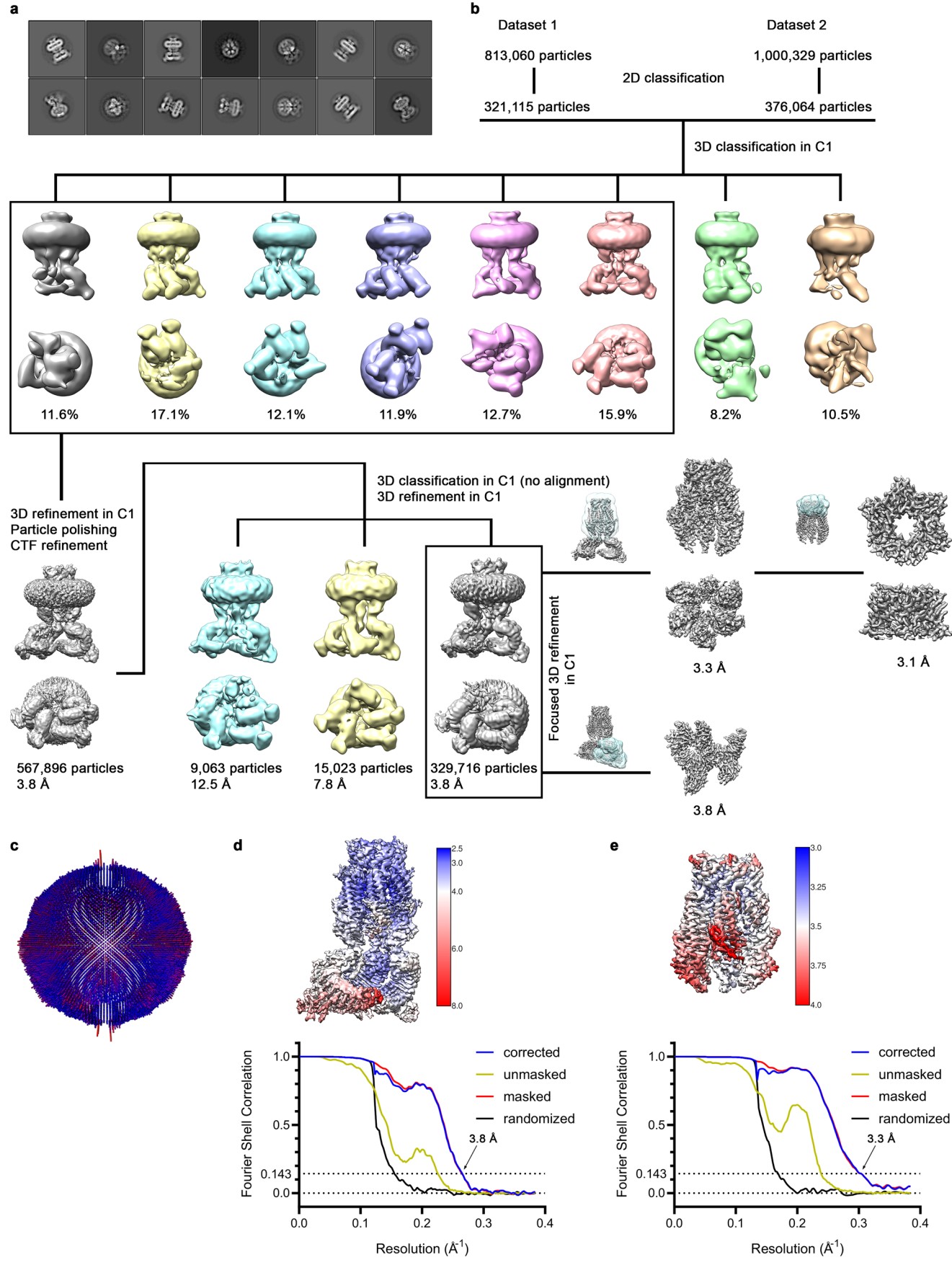

**Extended Data Fig. 4 | See next page for caption.**

**Extended Data Fig. 4 | Structure of the LRRC8A/C channel in complex with Sb1. a**, Representative 2D class averages of the LRRC8A/C[1:1]/Sb1 complex. **b**, Data processing workflow. To preserve the unique structural details, all 3D classification and refinement steps were done without applying symmetry in C1. Eight classes generated during 3D classification reveal a well-defined pore domain and structural heterogeneity in the LRRDs, which reflects their intrinsic mobility. Particles assigned to the boxed classes containing at least one ordered pair of LRRDs with Sb1 bound were used in further refinement. The distribution of particles (%) is indicated. 3D refinement using all selected particles as input resulted in a reconstruction with an overall resolution of 3.8 Å. To recover less abundant assemblies, multiple rounds of 3D classification without alignment step followed by 3D refinement were performed. By this approach, particles were assigned to three different classes, each showing unique assemblies. The focused refinement of the PD and the selectivity filter of the most abundant class improved its resolution to 3.3 Å and 3.1 Å, respectively. Focused refinement of a tightly interacting LRRC8A domain pair in complex with Sb1 strongly improved its density and increased the resolution of this part of the structure to 3.8 Å. The view of the domain pair is rotated by 60° with respect to the full-length refined reconstruction. Insets show the masked regions during refinement. **c**, Angular distribution plot of all particles included in the final reconstruction of the full-length complex. The length and color of cylinders correspond to the number of particles with respective Euler angles. **d, e**, Final 3D reconstruction colored according to local resolution (top) and FSC plot of the final refined unmasked (yellow), masked (red), phase-randomized (black) and corrected for mask convolution effects (blue) cryo-EM density map of the LRRC8A/C[1:1]/Sb1 complex. The resolution at which the FSC curve drops below the 0.143 threshold is indicated. **d**, Full-length complex and **e**, masked TM region.

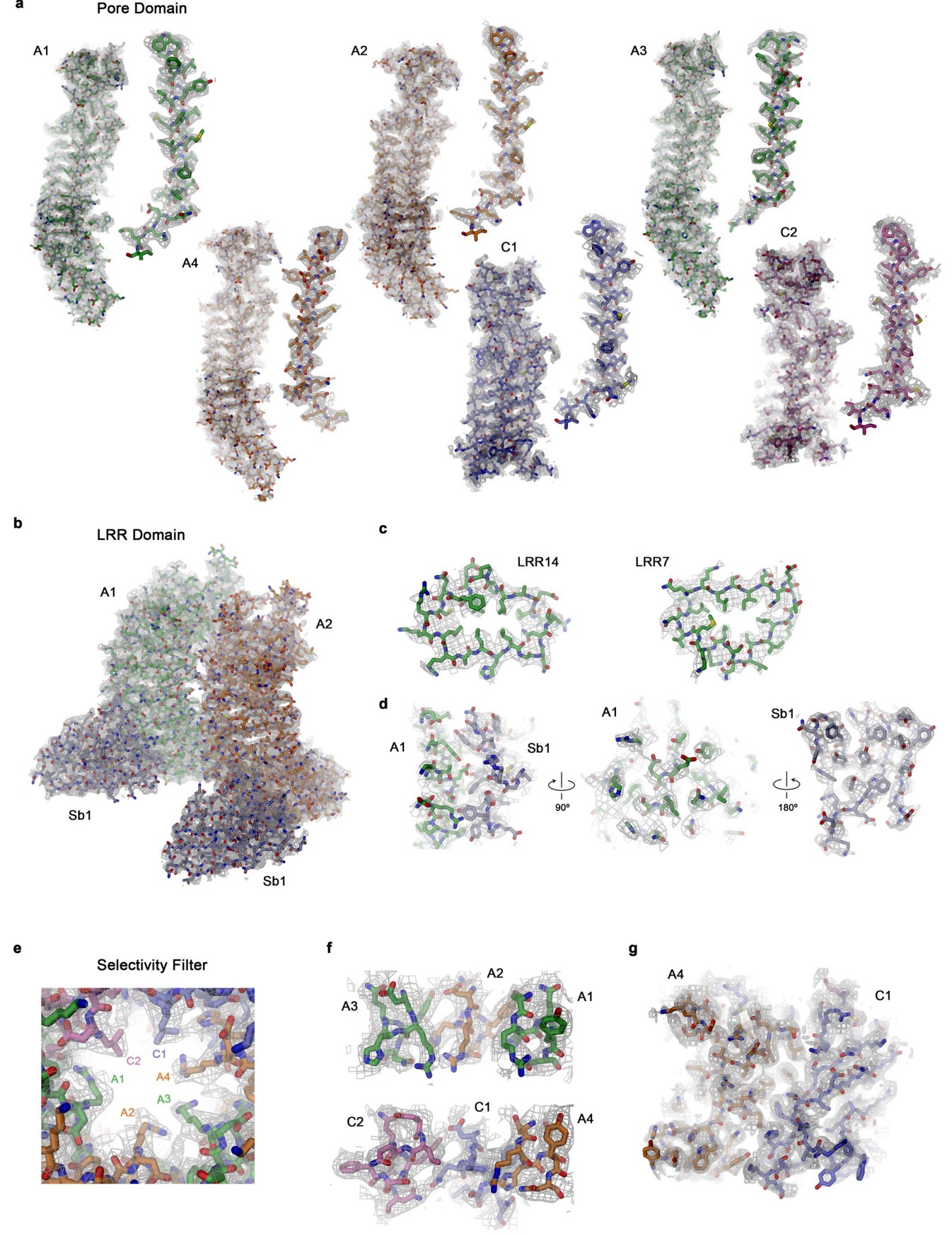

**Extended Data Fig. 5 | See next page for caption.**

**Extended Data Fig. 5 | Features in the cryo-EM map of the LRRC8A/C[1:1]/ Sb1 complex.** Sections of the cryo-EM density of the LRRC8A/C[1:1]/Sb1 dataset superimposed on the refined model. Subunits are labeled as in Fig. 3. **a**, Density of the PD at 3.3 Å superimposed on each of the six subunits. The PD of respective subunits is shown left, the first membrane-spanning α-helix TM1 right. The density is contoured at 12 σ for A1-A4 and at 8 σ for C1 and C2. **b-d**, Masked density around a pair of tightly interacting LRRC8A domains in complex with Sb1 at 3.8 Å (contoured at 15 σ). **b**, LRR domain pair with bound Sb1. **c**, Leucine rich repeats 14 (left) and 7 (right) of the LRRD of subunit A1. **d**, LRRD-Sb1 interaction interface (left) and open book representation with the LRRD of subunit A1 shown in the center and Sb1 on the right. **e-g**, Selectivity filter with cryo-EM density of the masked region at 3.1 Å (contoured at 12 σ) shown superimposed. **e**, Pore constriction viewed from the extracellular side. **f**, View of the inside of the filter parallel to the membrane. Top, subunits A1, A2, and A3, bottom, subunits A4, C1 and C2. **g**, Interaction interface of the ESD between subunits A4 and C1. **a-g**, The protein is shown as sticks, the cryo-EM density as gray mesh.

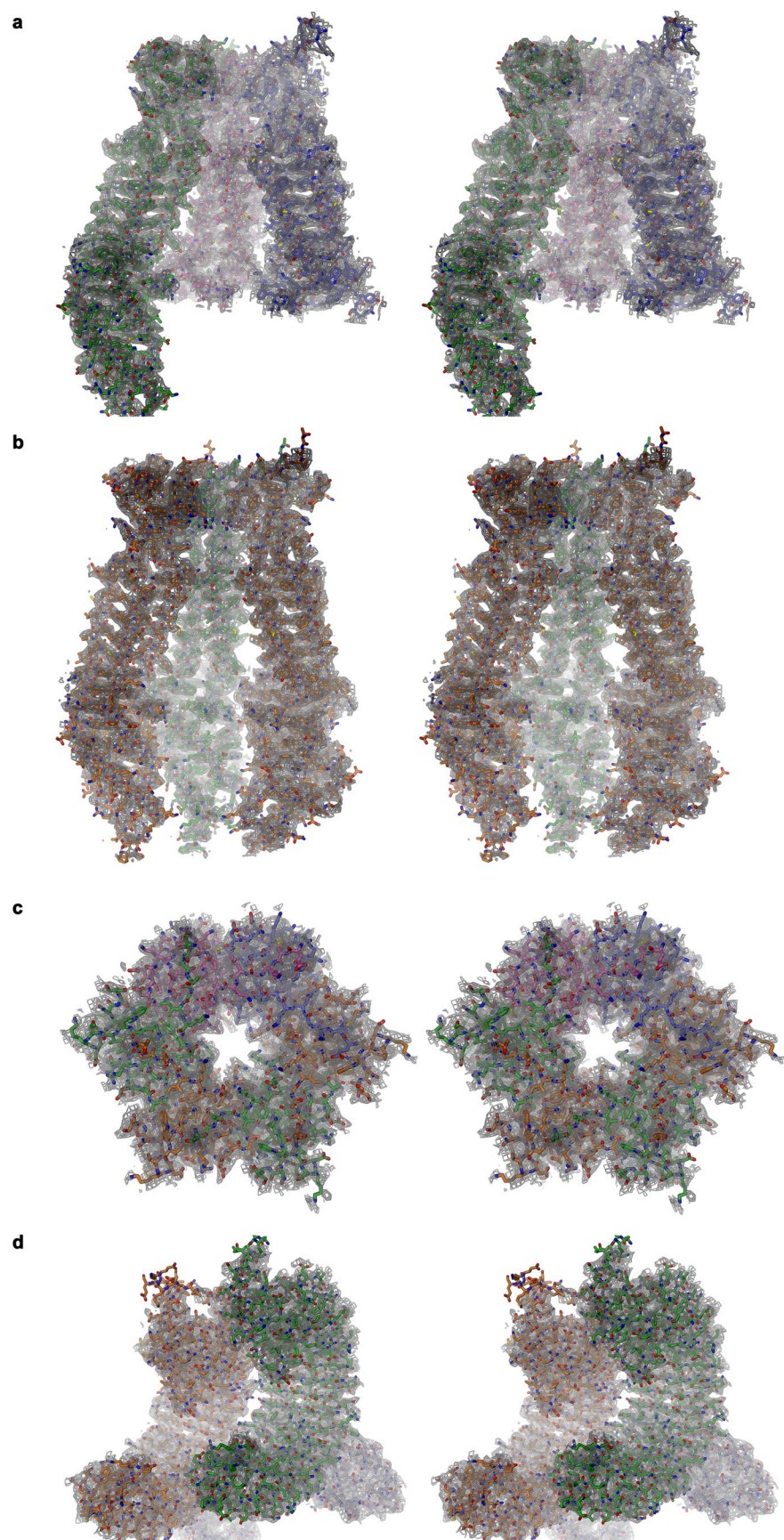

**Extended Data Fig. 6 | Stereo views of the LRRC8A/C^1:1/Sb1 complex.** Focused cryo-EM densities of the LRRC8A/C^1:1/Sb1 complex superimposed on different parts of the protein. **a**, **b**, Density of the PD at 3.3 Å. **a**, Subunits A1, C2 and C1, **b**, subunits A2, A3 and A4. **c**, Density of the selectivity filter at 3.1 Å viewed from the extracellular side. **d**, Tightly interacting LRRDs of A subunits in complex with Sb1 at 3.8 Å.

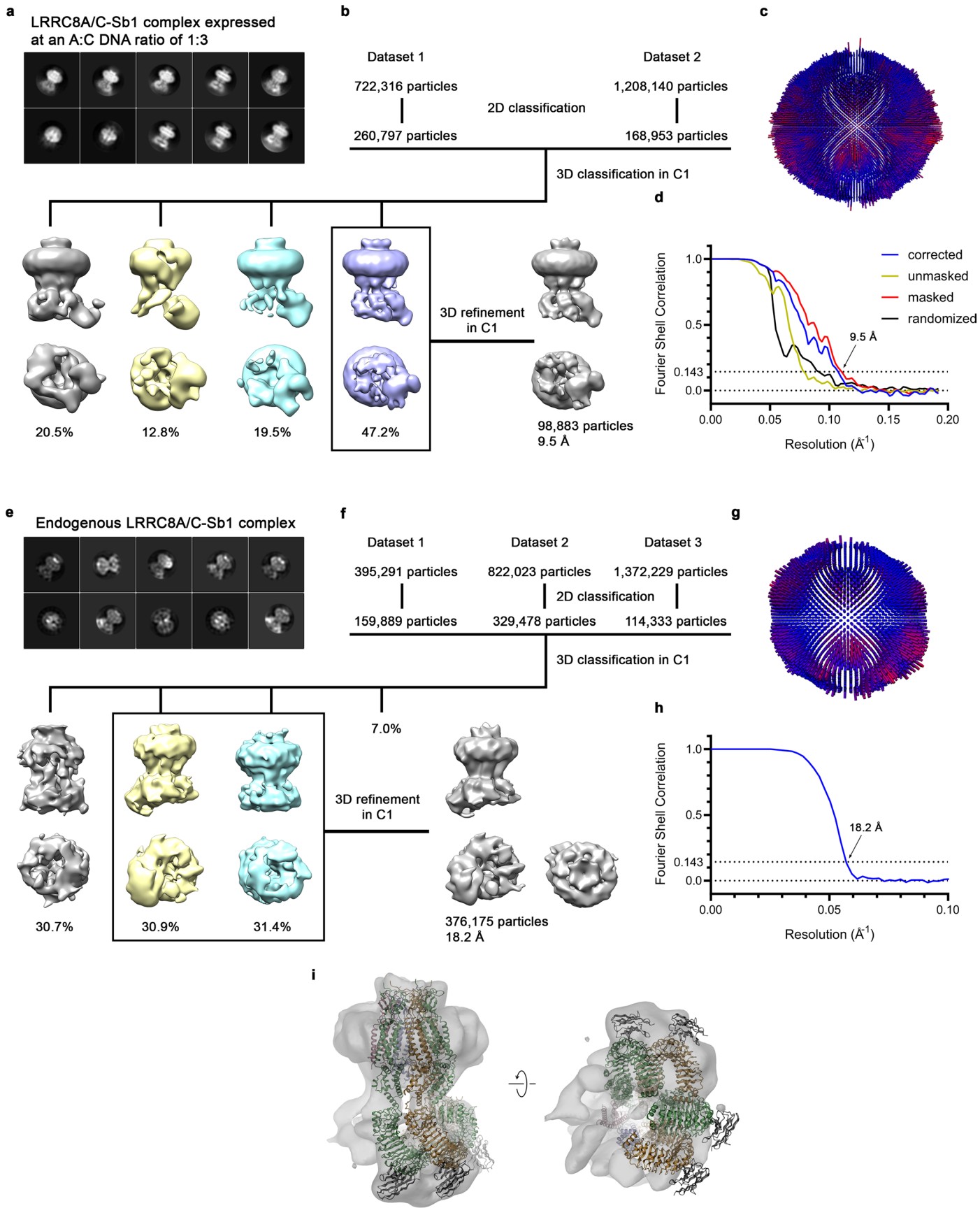

**Extended Data Fig. 7 | See next page for caption.**

**Extended Data Fig. 7 | Structure of overexpressed and endogenous LRRC8A/C/Sb1 complexes. a-d**, Structure of overexpressed channels obtained from a transfection with an LRRC8A:LRRC8C DNA ratio of 1:3. **a**, Representative 2D class averages of the LRRC8A/C[1:3]/Sb1 complex. **b**, Data processing workflow. To preserve the unique structural details, all 3D classification and refinement steps were carried out in C1. Three classes generated during 3D classification reveal a well-defined PD and pronounced structural heterogeneity in the LRRDs, compared to the sample of LRRC8A/C co-expressed at a 1:1 DNA ratio. Particles assigned to the boxed class containing a single ordered LRRD with Sb1 bound were used in further refinement. The distribution of particles (%) is indicated. 3D refinement using all selected particles as an input resulted in a reconstruction with an overall resolution of 9.5 Å. **c**, Angular distribution plot of all particles included in the final reconstruction of the full-length complex. The length and color of cylinders correspond to the number of particles with respective Euler angles. **d**, FSC plot of the final refined unmasked (yellow), masked (red), phase-randomized (black) and corrected for mask convolution effects (blue) cryo-EM density map of the LRRC8A/C[1:3]/Sb1 complex. The resolution at which the FSC curve drops below the 0.143 threshold is indicated. **e-i**, Structure of endogenous channels purified from LRRC8[B,D,E-/-] cells. **e**, Representative 2D class averages of the endogenous LRRC8A/C in complex with Sb1. **f**, Data processing workflow. To preserve the unique structural details, all 3D classification and refinement steps were carried out in C1. Three (displayed) out of four classes generated during 3D classification reveal a characteristic channel architecture, consisting of a comparably well resolved PD and heterogenous LRRDs. Particles assigned to the boxed classes, which show the best-defined overall architecture, were used in further refinement. The distribution of particles (%) is indicated. 3D refinement using all selected particles as an input resulted in a reconstruction with an overall resolution of 18.2 Å. **g**, Angular distribution plot of all particles included in the final reconstruction of the full-length complex. The length and color of cylinders correspond to the number of particles with respective Euler angles. **h**, FSC plot of the final refined cryo-EM density map of the endogenous LRRC8A/C[endog]/Sb1 complex. The resolution at which the FSC curve drops below the 0.143 threshold is indicated. **i**, Fit of the LRRC8A/C/Sb1 model (ribbon) into the cryo-EM density of the endogenous channel complex.

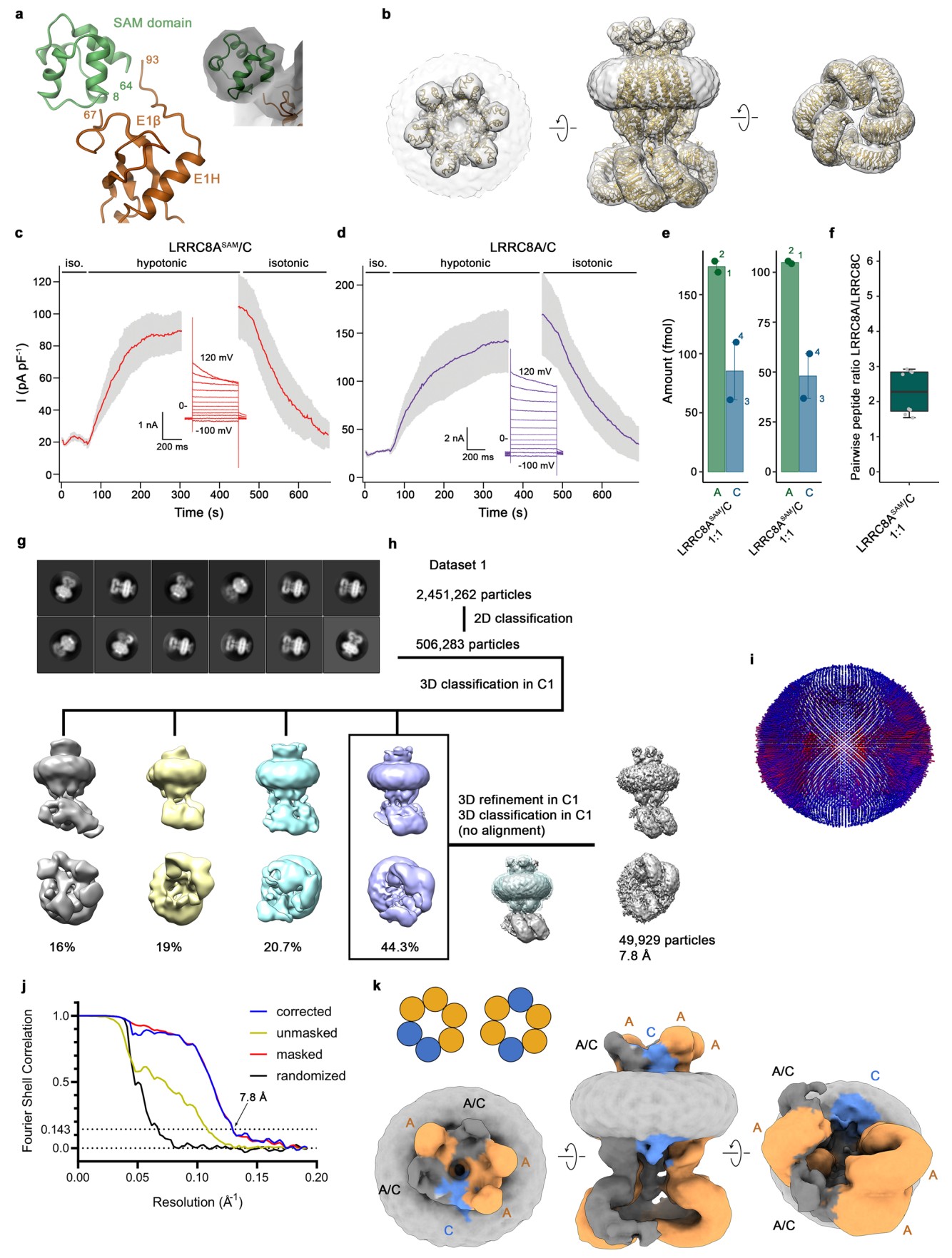

**Extended Data Fig. 8 | See next page for caption.**

**Extended Data Fig. 8 | Structure of channels containing a genetically modified LRRC8A subunit. a**, Close-up of a SAM domain fused to the truncated first extracellular loop of LRRC8A, connecting residue 67 at the end of the β-strand E1β with residue 93 preceding the α-helix E1H. The inset shows the SAM domain fitted into cryo-EM density of the homomeric LRRC8A$^{SAM}$ channel. **b**, Cryo-EM density of the homomeric LRRC8A$^{SAM}$ construct at 6.9 Å, showing a channel with similar conformational properties as observed in the structure of the unlabeled LRRC8A subunit. The views from within the membrane (center) and the extracellular side (left) show extra density corresponding to fused SAM domains for all subunits in the C3 symmetrized map. **c, d**, VRAC currents recorded from LRRC8$^{-/-}$ cells expressing LRRC8A/C constructs in the whole-cell configuration in response to a change of the extracellular environment to hypotonic conditions. Values show average of four biological replicates recorded at 100 mV for, **c**, LRRC8A$^{SAM}$/C and, **d**, LRRC8A/C. Errors are s.e.m. The changes in extracellular osmolarity are indicated (top). Insets show traces of a representative cell recorded at different voltages at the plateau of activation. **e**, Quantification of absolute peptide levels in isolated LRRC8A$^{SAM}$/C channels using LC-MS/MS. Peptide amounts on the column (fmol) were calculated using the ratio of endogenous and SIL peptides (as defined in Extended Data Fig. 1d and e). Data was obtained from two independent measurements of the same sample. **f**, Ratio determination of LRRC8A to LRRC8C in isolated complexes using LC-MS/MS. All pairwise ratios of LRRC8A peptides relative to LRRC8C peptides were obtained as described in Fig. 1b. Absolute peptide levels calculated by spiking each sample with known amounts of stable isotope-labeled peptides were used for ratio determination. Boxplots cover the first and third quartiles from bottom to top and the whisker extends to largest/smallest value but no further than 1.5×IQR (inter-quartile range). The median ratio is indicated by a black solid line. **g**, Representative 2D class averages of the LRRC8A$^{SAM}$/C

complex. **h**, Data processing workflow. To preserve the unique structural details, all 3D classification and refinement steps were carried out in C1. Four classes generated during 3D classification reveal a well-defined pore domain and structural heterogeneity in the LRRDs, which reflects their intrinsic mobility. Particles assigned to the boxed class with one ordered pair of LRRDs were used in further refinement. The distribution of particles (%) is indicated. 3D refinement using all selected particles as input resulted in a reconstruction with an overall resolution of 8.6 Å. As this reconstruction indicated that both populations of the LRRC8A$^{SAM}$/C complex with distinct subunit arrangement might be averaged, iterative focused 3D classification with a mask around the subunits with stable positions followed by 3D refinement was performed to separate these two populations. Despite the described efforts, the final refined map still contains an average of two assemblies. **i**, Angular distribution plot of all particles included in the final reconstruction of the full-length complex. The length and color of cylinders correspond to the number of particles with respective Euler angles. **j**, FSC plot of the final refined unmasked (yellow), masked (red), phase-randomized (black) and corrected for mask convolution effects (blue) cryo-EM density map of the LRRC8A$^{SAM}$/C complex. The resolution at which the FSC curve drops below the 0.143 threshold is indicated. **k**, Cryo-EM density of the LRRC8A$^{SAM}$/C complex showing a channel with two populations averaged in a consensus reconstruction as manifested in the density of the SAM domain and of the corresponding LRRDs of the respective subunits, both of which are strong for three positions that are only occupied by A subunits, intermediate in two positions where we find an average between A and C subunits and absent in one position that is only occupied by LRRC8C. Subunits are colored accordingly: A – orange, C – blue, mixture of A and C – gray. The insets show the schematic arrangement of A and C subunits in the two distinct channel assemblies.

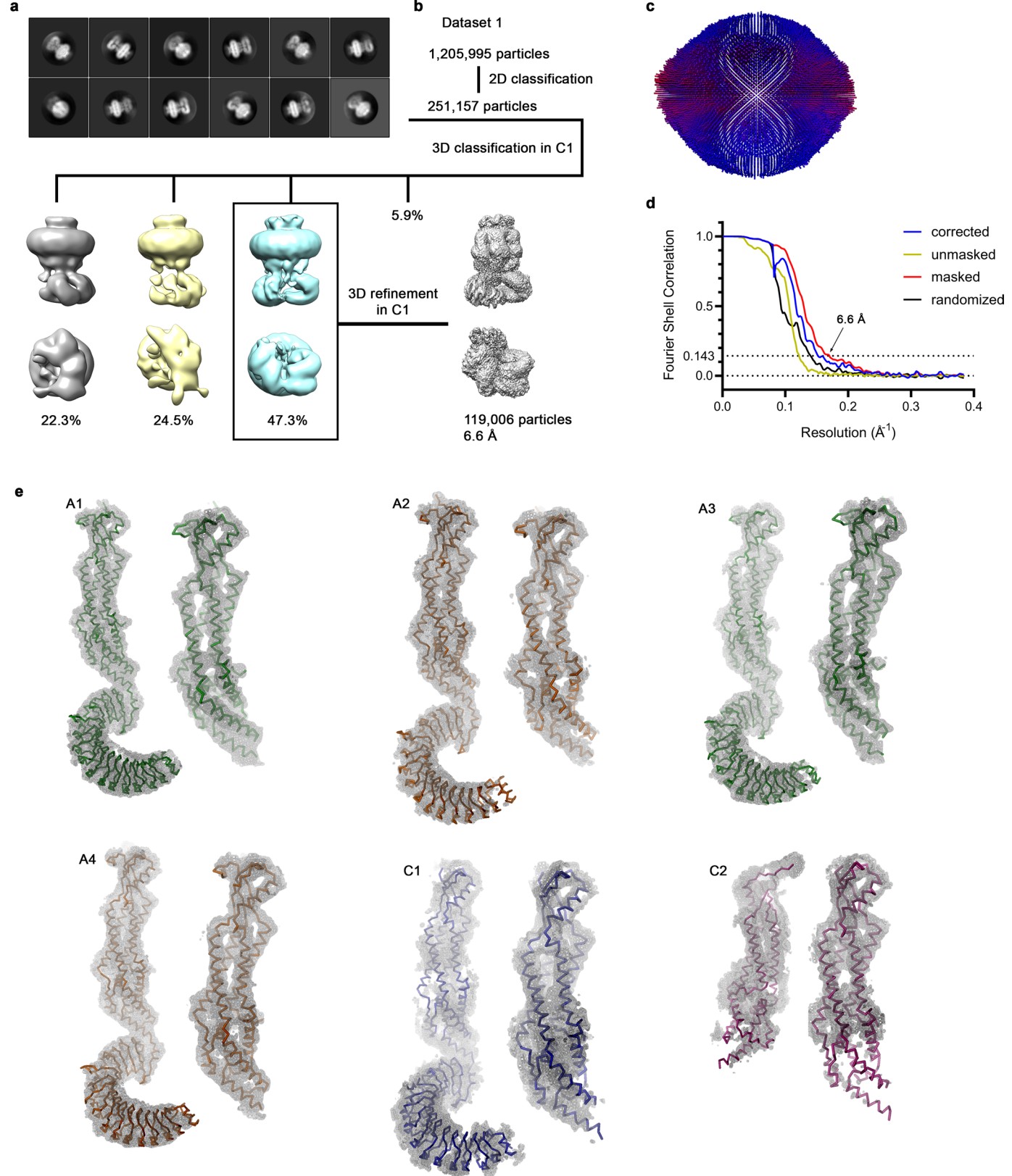

**Extended Data Fig. 9 | See next page for caption.**

**Extended Data Fig. 9 | Structure of LRRC8A/C channels in absence of subunit-specific labels. a**, Representative 2D class averages of the LRRC8A/C complex. **b**, Data processing workflow. To preserve the unique structural details, all 3D classification and refinement steps were carried out in C1. Three (displayed) out of four generated classes during 3D classification reveal a well-defined PD and structural heterogeneity in the LRRDs, reflecting their intrinsic mobility. Particles assigned to the boxed class with two ordered pairs of LRRDs were used in further refinement steps. The distribution of particles (%) is indicated. 3D refinement using all selected particles as input resulted in a reconstruction with an overall resolution of 6.6 Å. **c**, Angular distribution plot of all particles included in the final reconstruction of the full-length complex. The length and color of cylinders correspond to the number of particles with respective Euler angles. **d**, FSC plot of the final refined unmasked (yellow), masked (red), phase-randomized (black) and corrected for mask convolution effects (blue) cryo-EM density map of the LRRC8A/C complex. The resolution at which the FSC curve drops below the 0.143 threshold is indicated. **e**, Cryo-EM density of the LRRC8A/C complex at 6.6 Å superimposed on a Cα-representation of the six individual subunits (labeled as in Fig. 3a). For each subunit, the maps are shown at lower contour (A subunits at 10 σ, C subunits at 6 σ) for the entire protein (left) and at higher contour (A subunits at 12 σ, C subunits at 10 σ) for the PD (right).

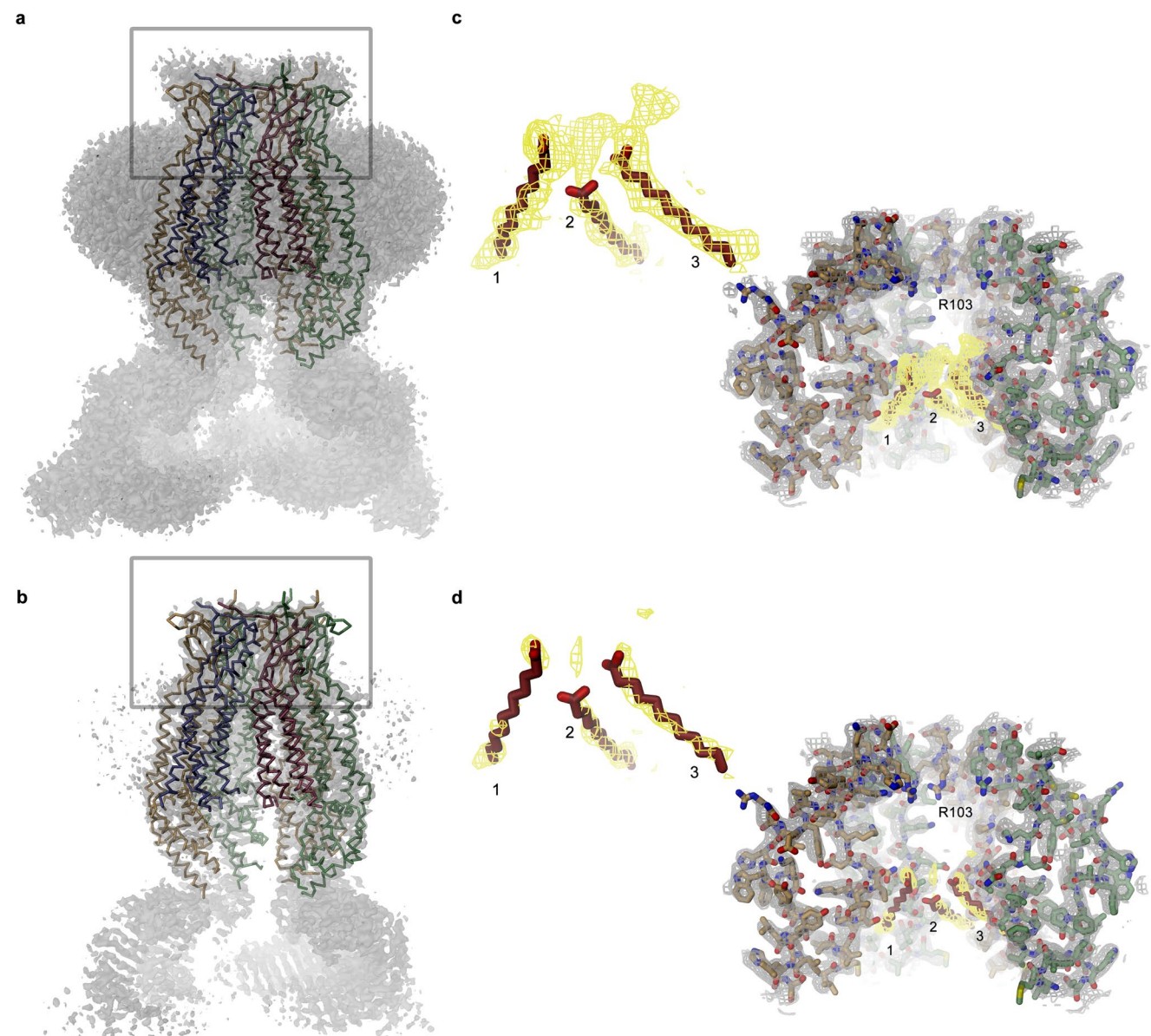

**Extended Data Fig. 10 | Residual density in the pore of the LRRC8A/C[1:1]/ Sb1 complex. a, b**, Cryo-EM density at 3.8 Å of the entire LRRC8A/C[1:1]/Sb1 complex contoured at 7σ (**a**) and 13σ (**b**) superimposed on the Cα model of the PD. Frame indicates regions displayed in the following panels. **c, d**, Zoom into the extracellular part of the PD. The same cryo-EM density as shown in panel **a** contoured at 7σ (**c**) and 13σ (**d**) is superimposed on a stick model of the four A subunits. The view is from within the pore, the orientation is as in panel **a**. Residual density in the pore region is shown in yellow with fatty acid chains placed as reference. Blow up of the same density is shown left. Equivalent positions are numbered and the location of Arg 103 at the selectivity filter is labeled.

Marta Sawicka

# Reporting Summary

Nature Research wishes to improve the reproducibility of the work that we publish. This form provides structure for consistency and transparency in reporting. For further information on Nature Research policies, see our Editorial Policies and the Editorial Policy Checklist.

## Statistics

For all statistical analyses, confirm that the following items are present in the figure legend, table legend, main text, or Methods section.

| n/a | Confirmed | |
|---|---|---|
| ☐ | ☒ | The exact sample size ($n$) for each experimental group/condition, given as a discrete number and unit of measurement |
| ☐ | ☒ | A statement on whether measurements were taken from distinct samples or whether the same sample was measured repeatedly |
| ☒ | ☐ | The statistical test(s) used AND whether they are one- or two-sided<br>*Only common tests should be described solely by name; describe more complex techniques in the Methods section.* |
| ☒ | ☐ | A description of all covariates tested |
| ☒ | ☐ | A description of any assumptions or corrections, such as tests of normality and adjustment for multiple comparisons |
| ☐ | ☒ | A full description of the statistical parameters including central tendency (e.g. means) or other basic estimates (e.g. regression coefficient) AND variation (e.g. standard deviation) or associated estimates of uncertainty (e.g. confidence intervals) |
| ☒ | ☐ | For null hypothesis testing, the test statistic (e.g. $F$, $t$, $r$) with confidence intervals, effect sizes, degrees of freedom and $P$ value noted<br>*Give P values as exact values whenever suitable.* |
| ☒ | ☐ | For Bayesian analysis, information on the choice of priors and Markov chain Monte Carlo settings |
| ☒ | ☐ | For hierarchical and complex designs, identification of the appropriate level for tests and full reporting of outcomes |
| ☒ | ☐ | Estimates of effect sizes (e.g. Cohen's $d$, Pearson's $r$), indicating how they were calculated |

*Our web collection on statistics for biologists contains articles on many of the points above.*

## Software and code

Policy information about availability of computer code

| Data collection | Clampex 10.6, EPU 2.9 |
|---|---|
| Data analysis | Clampfit 10.6, Excel 2108, CTFFIND 4.1, RELION 3.1.2 and RELION 4.0-beta, Phenix 1.20.1, Coot 0.9.8, DINO 0.9.6, Chimera 1.15, ChimeraX 1.2.5, HOLE 2.2.005, MSMS, Proteome Discoverer 2.1, Matrix Science 2.7 and 2.7.0.1, Scaffold (Proteome Software Inc. 5.10), XDS, Phaser |

For manuscripts utilizing custom algorithms or software that are central to the research but not yet described in published literature, software must be made available to editors and reviewers. We strongly encourage code deposition in a community repository (e.g. GitHub). See the Nature Research guidelines for submitting code & software for further information.

## Data

Policy information about availability of data

All manuscripts must include a data availability statement. This statement should provide the following information, where applicable:
- Accession codes, unique identifiers, or web links for publicly available datasets
- A list of figures that have associated raw data
- A description of any restrictions on data availability

The three-dimensional cryo-EM density maps have been deposited in the Electron Microscopy Data Bank under accession numbers EMD-15835 (LRRC8C), EMD-15836 (LRRC8A/C1:1/Sb1), EMD-15837 (LRRC8A/C), EMD-15838 (LRRC8A/C1:3/Sb1), EMD-15839 (LRRC8A/Cendog/Sb1), EMD-15840 (LRRC8ASAM), EMD-15841 (LRRC8ASAM/C). The deposition includes maps of full-length proteins, corresponding both half-maps, the mask used for final FSC calculation as well as relevant higher resolution maps obtained after local refinement. Coordinates have been deposited in the Protein Data Bank under accession numbers 8B40 (LRRC8C), 8B41 (LRRC8A/C1:1/Sb1), 8B42 (LRRC8A/C). Coordinates and structure factors of the X-ray structure of the LRRD of LRRC8C have been deposited in the PDB under accession number 8BEN. The mass spectrometry proteomics data have been deposited to the ProteomeXchange Consortium via the PRIDE (http://

# Field-specific reporting

Please select the one below that is the best fit for your research. If you are not sure, read the appropriate sections before making your selection.

☒ Life sciences ☐ Behavioural & social sciences ☐ Ecological, evolutionary & environmental sciences

For a reference copy of the document with all sections, see nature.com/documents/nr-reporting-summary-flat.pdf

# Life sciences study design

All studies must disclose on these points even when the disclosure is negative.

| | |
|---|---|
| Sample size | No sample size determination was performed. Quantification and functional experiments were performed multiple times with similar results and addition of further data did not change the conclusions of the study. Complete cryo-EM statistics are provided in Table 1 and 2 and in Extended Data Figures 3-4 and 7-9 |
| Data exclusions | In electrophysiology experiments, leaky recordings were discarded. Recordings with no current response were excluded from the analysis. 85% of wild type cells and 35% of LRRC8B,D,E-/- cells showed current response. HEK293 LRRC8-/- cells showed current response in 90% and 70% of patched cells when transfectedwith LRRC8A and LRRC8C or LRRC8ASAM and LRRC8C, respectively. |
| Replication | Mass spectrometry experiments were replicated as indicated in Extended Data Figure 1 and in the Method section, all replications were successful. Electrophysiology data show the mean of the indicated number of biological replicates, errors are indicated. Recordings were performed multiple times from different transfections. Cells without current response were excluded from the analysis (as described under data exclusion). |
| Randomization | Randomization is not relevant for this study, as there were no groups allocated in any of the experiments. |
| Blinding | No blinding was applied as this is deemed not practically feasible. |

# Reporting for specific materials, systems and methods

We require information from authors about some types of materials, experimental systems and methods used in many studies. Here, indicate whether each material, system or method listed is relevant to your study. If you are not sure if a list item applies to your research, read the appropriate section before selecting a response.

## Materials & experimental systems

| n/a | Involved in the study |
|---|---|
| ☐ | ☒ Antibodies |
| ☐ | ☒ Eukaryotic cell lines |
| ☒ | ☐ Palaeontology and archaeology |
| ☒ | ☐ Animals and other organisms |
| ☒ | ☐ Human research participants |
| ☒ | ☐ Clinical data |
| ☒ | ☐ Dual use research of concern |

## Methods

| n/a | Involved in the study |
|---|---|
| ☒ | ☐ ChIP-seq |
| ☒ | ☐ Flow cytometry |
| ☒ | ☐ MRI-based neuroimaging |

## Antibodies

| | |
|---|---|
| Antibodies used | The antibody used in this study (monoclonal anti-LRRC8A antibody produced in mouse) is commercially available (Sigma, SAB1412855, clone 8H9). |
| Validation | The mouse anti-LRRC8A antibody (Sigma, SAB1412855) was validated by the supplier and the validation report is available from their website. The antibody was additionally verified by Western Blot using purified LRRC8A protein (Deneka D. et al., Allosteric modulation of LRRC8 channels by targeting their cytoplasmic domains. Nat. Commun., doi: 10.1038/s41467-021-25742-w (2021)) |

## Eukaryotic cell lines

Policy information about cell lines

| | |
|---|---|
| Cell line source(s) | HEK293S GnTI- (CRL-3022) and HEK293T (CRL-1573) cells were obtained from ATCC. HEK293 LRRC8-/- and HEK293 LRRC8B,D,E-/- cells were obtained from the laboratory of T. J. Jentsch |
| Authentication | No further authentification was performed for the commercially available celllines. The lack of expression of LRRC8 proteins |

| Authentication | in HEK293 LRRC8-/- cells was confirmed by electrophysiligy and Western blots with anti-LRRC8A antibody. The expression of LRRC8A proteins in HEK293 LRRC8B,D,E-/- cells was confirmed by Western blots with anti-LRRC8A antibody. |
| --- | --- |
| Mycoplasma contamination | The cell lines were tested and are free from mycoplasma contamination. |
| Commonly misidentified lines (See ICLAC register) | No commonly misidentified lines were used in the study. |

