## [Peer Review File · Nature Structural & Molecular Biology]

Peer Review Information

Manuscript Title: Structure of a volume-regulated heteromeric LRRC8A/C channel

Corresponding author name(s): Marta Sawicka, Raimund Dutzler

Reviewer Comments & Decisions:

Decision Letter, initial version:
--

Message: 30th Sep 2022

Dear Raimund,

Thank you again for submitting your manuscript "Structure of a volume-regulated heteromeric LRRC8A/C channel". We now have comments (below) from the 2 reviewers who evaluated your paper. In light of those reports, we remain very interested in your study and would like to see your response to the comments of the referees, in the form of a revised manuscript.

I hope you will be pleased to see that the reviewers are very positive about the quality and interest of the work. Nevertheless, they request a few additional experiments and textual changes. Please be sure to address/respond to all concerns of the referees in full in a point-by-point response and highlight all changes in the revised manuscript text file. If you have comments that are intended for editors only, please include those in a separate cover letter.

We expect to see your revised manuscript within 6 weeks. If you cannot send it within this time, please contact us to discuss an extension; we would still consider your revision, provided that no similar work has been accepted for publication at NSMB or published elsewhere.

As you already know, we put great emphasis on ensuring that the methods and statistics

reported in our papers are correct and accurate. As such, if there are any changes that should be reported, please submit an updated version of the Reporting Summary along with your revision.

Reporting Summary:

When submitting the revised version of your manuscript, please pay close attention to our [Digital Image Integrity Guidelines](https://www.nature.com/nature-portfolio/editorial-policies/image-integrity).

Data availability: this journal strongly supports public availability of data. All data used in accepted papers should be available via a public data repository, or alternatively, as Supplementary Information. If data can only be shared on request, please explain why in your Data Availability Statement, and also in the correspondence with your editor. Please note that for some data types, deposition in a public repository is mandatory - more information on our data deposition policies and available repositories can be found below: <https://www.nature.com/nature-research/editorial-policies/reporting-standards#availability-of-data>

We require deposition of coordinates (and, in the case of crystal structures, structure factors) into the Protein Data Bank with the designation of immediate release upon publication (HPUB). Electron microscopy-derived density maps and coordinate data must be deposited in EMDB and released upon publication. To avoid delays in publication, dataset accession numbers must be supplied with the final accepted manuscript and appropriate release dates must be indicated at the galley proof stage.

While we encourage the use of color in preparing figures, please note that this will incur a charge to partially defray the cost of printing. Information about color charges can be found at <https://www.nature.com/nsmb/authors/submit/index.html#costs>

[Redacted]

Kind regards,
Florian

Dr Florian Ullrich
Associate Editor, Nature
Consulting Editor, Nature Structural & Molecular Biology
ORCID 0000-0002-1153-2040

Referee expertise:

Referee #1: cryo-EM, channels

Referee #2: VRAC function

Reviewers' Comments:

Reviewer #1:

Remarks to the Author:

LRRc8 proteins form volume regulated anion channels (VRAC). Physiological LRRc8 channels are heteromeric assemblies, which consist of the obligatory LRRc8A subunit and additional (LRRc8B-E) subunits. The composition and stoichiometry of physiological VRAC assemblies remain unknown. In this manuscript, the authors report cryo-EM structures of heteromeric LRRc8A:C channels. Overall, the structural studies are quite comprehensive and carefully executed. Interesting structural findings include the 2:1 A:C stoichiometry,

heptameric LRRC8C homomers, and alternative subunit arrangement of the SAM domain fused LRRC8A/C heteromer. Overall, the results are well presented, the analyses are well supported, and the manuscript is well written. This work represents an important advance to our understanding of physiological LRRC8 channels. I only have several minor points.

Abstract: "Here we have closed this gap..." is overstated because other heteromeric structures/compositions may differ from LRRC8A/C.

Line 57: "low-resolution reconstitution", should be low-resolution reconstruction.

The LRRC8A(SAM)/C complex has an altered subunit disposition compared with the A/C heteromer. The authors think that it is likely a consequence of the fused SAM domain. Is the LRRC8A(SAM)/C channel functional? This is an important point worth further discussion because fusion strategy is often used in structural biology. Perhaps the authors can elaborate more about potential artifacts with fusion.

Interestingly, another preprint also reported structures of the LRRC8A/C heteromer, but with very different stoichiometry (5:1 A:C complex). Additionally, the other preprint reported lipid-like densities in the pore and suggested that lipids may be involved in channel gating. Did the authors also observe similar lipid densities? Could the authors use their comprehensive structural findings to compare with the other study? This will substantially benefit researchers in the field regarding these contrasting results.

Reference:

Structural basis for assembly and lipid-mediated gating of LRRC8A:C volume-regulated anion channels

David M. Kern, Julia Bleier, Somnath Mukherjee, Jennifer M. Hill, Anthony A. Kossiakoff, Ehud Y. Isacoff, Stephen G. Brohawn doi: <https://doi.org/10.1101/2022.07.31.502239>

Reviewer #2:

Remarks to the Author:

Volume-regulated anion channel (VRAC) plays a key role in cell volume regulation. In addition, this ubiquitous channel is also important for many other physiological and pathological processes. After several decades of searching, its molecular identity was finally discovered in 2014. The channel is formed by the obligatory subunit LRRC8A (also known as SWELL1) and at least one of its paralogs (LRRC8B-E). The subunit composition is thought to regulate channel properties. Subsequently, cryo-EM structures of LRRC8A (and LRRC8D) homomeric channels were determined by several groups including the authors of this current manuscript. Collectively, these studies establish the basic architecture of VRAC. Two major (and challenging) questions remain in the field: 1) the native heterometric VRAC structure (subunit stoichiometry and their arrangement); and how different compositions determine various channel properties (especially substrate specificity); 2) how cell swelling (and other stimuli) activates/gates the channel.

This current manuscript by Sonja Rutz et al represents a major step in answering the first question and provides the basis for exploring the remaining puzzles in the field. I read the paper with great interest and find the data to be compelling and thorough. The authors

employed multiple innovative methods (MS, X-ray, cryo-EM) and expression systems (native, varying cDNA ratio, sybody, SAM fiducial tag) to address the complicated question on the stoichiometry and structure of heteromeric channels. The data quality is very high and conclusions well supported. The paper is also really well written and the figures are clear. I have only one suggestion for the revision. The authors may want to provide electrophysiological data to support the LRRC8A SAM construct used for the purification indeed generates functional VRAC currents. Overall, I consider this to be an excellent paper that will be a landmark in the field and appreciated by the wide readership of NSMB. Congrats to the authors.

Author Rebuttal to Initial comments

We thank the reviewers for their constructive comments, which we have addressed in detail below and which we considered in our revision. Our revised manuscript now contains electrophysiology data that demonstrate that the LRRC8A^{SAM} construct is functional (Extended Data Fig. 8c, d) and a brief discussion of residual lipid-like density within the pore (Extended Data Fig. 10). We have also adjusted our manuscript to conform to the format of Nature Structural and Molecular Biology. The abstract is 140 words long and the main text was shortened to 4,492 words. The manuscript now contains eight display items (six figures and two tables), ten extended data figures, and three supplementary videos.

Response to reviewer's comments:

Reviewer #1:

Remarks to the Author:

LRRC8 proteins form volume regulated anion channels (VRAC). Physiological LRRC8 channels are heteromeric assemblies, which consist of the obligatory LRRC8A subunit and additional (LRRC8B-E) subunits. The composition and stoichiometry of physiological VRAC assemblies remain unknown. In this manuscript, the authors report cryo-EM structures of heteromeric LRRC8A:C channels. Overall, the structural studies are quite comprehensive and carefully executed. Interesting structural findings include the 2:1 A:C stoichiometry,

heptameric LRRC8C homomers, and alternative subunit arrangement of the SAM domain fused LRRC8A/C heteromer. Overall, the results are well presented, the analyses are well supported, and the manuscript is well written. This work represents an important advance to our understanding of physiological LRRC8 channels.

We thank the reviewer for the supporting comments.

I only have several minor points.

Abstract: “Here we have closed this gap...” is overstated because other heteromeric structures/compositions may differ from LRRC8A/C.

We have reworded the abstract to:

Line 14:

‘Here we have addressed this question by the structural characterization of LRRC8A/C channels.’

Line 57: “low-resolution reconstitution”, should be low-resolution reconstruction.

We have introduced the correction.

The LRRC8A(SAM)/C complex has an altered subunit disposition compared with the A/C heteromer. The authors think that it is likely a consequence of the fused SAM domain. Is the LRRC8A(SAM)/C channel functional?

We have characterized the LRRC8A^{SAM}/C channel by electrophysiology and found it to be functional. The data is shown in Extended Data Fig. 8c, d.

This is an important point worth further discussion because fusion strategy is often used in structural biology. Perhaps the authors can elaborate more about potential artifacts with fusion.

We have modified the sentences in line 222-238:

‘The altered subunit disposition in this channel population is likely a consequence of the fused SAM domain, which appears to mildly perturb the interaction between LRRC8A subunits, leading to the dissociation of contacts at the loose interface. Although these properties illustrate that even a considerate modification of the expression construct might affect the channel assembly, the preserved 2:1 A to C stoichiometry and the pairwise organization of tightly interacting LRRC8A subunits further support their role as building blocks in heteromeric VRACs.’

In light of the size limit of Nature Structural Biology (4,500 words maximum), which required us to shorten the manuscript we prefer not to engage in an extended discussion, which would also distract from the main focus of our work.

Interestingly, another preprint also reported structures of the LRRC8A/C heteromer, but with very different stoichiometry (5:1 A:C complex). Additionally, the other preprint reported lipid-like densities in the pore and suggested that lipids may be involved in channel gating. Did the authors also observe similar lipid densities? Could the authors use their comprehensive structural findings to compare with the other study? This will substantially benefit researchers in the field regarding these contrasting results.

Reference:

Structural basis for assembly and lipid-mediated gating of LRRC8A:C volume-regulated anion channels David M. Kern, Julia Bleier, Somnath Mukherjee, Jennifer M. Hill, Anthony A. Kossiakoff, Ehud Y. Isacoff, Stephen G. Brohawn doi:

<https://doi.org/10.1101/2022.07.31.502239>

We have referred to the study at two places in the discussion:

Line 300-304:

‘A 5:1 ratio of A to C subunits was reported in a recent structural study of a heteromeric complex containing a genetically modified fusion construct of LRRC8A³¹, resembling the approach with the LRRC8A^{SAM} fusion used here. The nature of this discrepancy is currently unclear and could be either a consequence of the used construct or related to the different expression host.’

Line 338-341:

‘A recent study has proposed a role of pore-lining lipids in channel gating based on residual density at the extracellular part of the TMD³¹. Although similar weak density is found in the LRRC8A/C^{1:1}/Sb1 complex, it is not sufficiently detailed to warrant such conclusion (Extended Data Fig. 10). A potential role of lipids in VRAC gating thus requires further investigation.’

As documented in Extended Data Fig. 10, we do find similar density in the pore region at low contour of the map that is somewhat above the level of the density of the detergent belt. Although its elongated shape is consistent with acyl chains, there are no features distinguishing it clearly as phospholipids. Though we find the proposal that lipids might be involved in the gating of VRACs clearly interesting, we do not think that our data provides conclusive evidence for such mechanism.

Reviewer #2:

Remarks to the Author:

Volume-regulated anion channel (VRAC) plays a key role in cell volume regulation. In addition,

this ubiquitous channel is also important for many other physiological and pathological processes. After several decades of searching, its molecular identity was finally discovered in 2014. The channel is formed by the obligatory subunit LRRC8A (also known as SWELL1) and at least one of its paralogs (LRRC8B-E). The subunit composition is thought to regulate channel properties. Subsequently, cryo-EM structures of LRRC8A (and LRRC8D) homomeric channels were determined by several groups including the authors of this current manuscript. Collectively, these studies establish the basic architecture of VRAC. Two major (and challenging) questions remain in the field: 1) the native heteromeric VRAC structure (subunit stoichiometry and their arrangement); and how different compositions determine various channel properties (especially substrate specificity); 2) how cell swelling (and other stimuli) activates/gates the channel.

This current manuscript by Sonja Rutz et al represents a major step in answering the first question and provides the basis for exploring the remaining puzzles in the field. I read the paper with great interest and find the data to be compelling and thorough. The authors employed multiple innovative methods (MS, X-ray, cryo-EM) and expression systems (native, varying cDNA ratio, sybody, SAM fiducial tag) to address the complicated question on the stoichiometry and structure of heteromeric channels. The data quality is very high and conclusions well supported. The paper is also really well written and the figures are clear.

We thank the reviewer for the supporting comments.

I have only one suggestion for the revision. The authors may want to provide electrophysiological data to support the LRRC8A SAM construct used for the purification indeed generates functional VRAC currents.

We now provide such data in Extended Data Fig. 8c, d.

Overall, I consider this to be an excellent paper that will be a landmark in the field and appreciated by the wide readership of NSMB. Congrats to the authors.

Decision Letter, first revision:

Message: Our ref: NSMB-A46698A

7th Nov 2022

Dear Raimund,

Thank you for submitting your revised manuscript "Structure of a volume-regulated heteromeric LRRC8A/C channel" (NSMB-A46698A). It has now been seen by the original referees and their comments are below. I hope you will be pleased to see that the reviewers find that the paper has improved in revision. Therefore we'll be happy in principle to publish it in Nature Structural & Molecular Biology, pending minor revisions to satisfy the referees' final requests and to comply with our editorial and formatting guidelines.

Kind regards,
Florian

Dr Florian Ullrich
Associate Editor, Nature
Consulting Editor, Nature Structural & Molecular Biology
ORCID 0000-0002-1153-2040

Reviewer #1 (Remarks to the Author):

The authors have very nicely addressed all my concerns and I recommend it's acceptance.

Reviewer #2 (Remarks to the Author):

Thanks for the new data showing the fusion construct is functional. Since SWELL1 (another name for LRRC8A) is also widely used for VRAC studies in the literature, the authors may consider refer it once in the abstract to increase the cross searchability.

Decision letter, author guidance:

Message: Our ref: NSMB-A46698A

8th Nov 2022

Dear Dr. Dutzler,

Thank you for your patience as we've prepared the guidelines for final submission of your Nature Structural & Molecular Biology manuscript, "Structure of a volume-regulated heteromeric LRRC8A/C channel" (NSMB-A46698A). Please carefully follow the step-by-step instructions provided in the attached file, and add a response in each row of the table to indicate the changes that you have made. Please also check and comment on any additional marked-up edits we have proposed within the text. Ensuring that each point is addressed will help to ensure that your revised manuscript can be swiftly handed over to our production team.

We would like to start working on your revised paper, with all of the requested files and forms, as soon as possible. We have fast-tracked your manuscript for publication as soon as possible, and would be grateful if you for the quickest turn around of revisions you can provide.

In recognition of the time and expertise our reviewers provide to Nature Structural & Molecular Biology's editorial process, we would like to formally acknowledge their contribution to the external peer review of your manuscript entitled "Structure of a volume-regulated heteromeric LRRC8A/C channel". For those reviewers who give their assent, we will be publishing their names alongside the published article.

Nature Structural & Molecular Biology offers a Transparent Peer Review option for new original research manuscripts submitted after December 1st, 2019. As part of this initiative, we encourage our authors to support increased transparency into the peer

review process by agreeing to have the reviewer comments, author rebuttal letters, and editorial decision letters published as a Supplementary item. When you submit your final files please clearly state in your cover letter whether or not you would like to participate in this initiative. Please note that failure to state your preference will result in delays in accepting your manuscript for publication.

Cover suggestions

As you prepare your final files we encourage you to consider whether you have any images or illustrations that may be appropriate for use on the cover of Nature Structural & Molecular Biology.

Nature Structural & Molecular Biology has now transitioned to a unified Rights Collection system which will allow our Author Services team to quickly and easily collect the rights and permissions required to publish your work. Approximately 10 days after your paper is formally accepted, you will receive an email in providing you with a link to complete the grant of rights. If your paper is eligible for Open Access, our Author Services team will also be in touch regarding any additional information that may be required to arrange payment for your article.

Please note that *Nature Structural & Molecular Biology* is a Transformative Journal (TJ). Authors may publish their research with us through the traditional subscription access route or make their paper immediately open access through payment of an article-processing charge (APC). Authors will not be required to make a final decision about access to their article until it has been accepted. [Find out more about Transformative Journals](https://www.springernature.com/gp/open-research/transformative-journals)

Authors may need to take specific actions to achieve [compliance with funder and institutional open access mandates](https://www.springernature.com/gp/open-research/funding/policy-compliance-faqs). If your research is supported by a funder that requires immediate open access (e.g. according to [Plan S principles](https://www.springernature.com/gp/open-research/plan-s-compliance)) then you should select the gold OA route, and we will direct you to the compliant route where possible. For authors selecting the subscription

publication route, the journal's standard licensing terms will need to be accepted, including [self-archiving policies](https://www.nature.com/nature-portfolio/editorial-policies/self-archiving-and-license-to-publish). Those licensing terms will supersede any other terms that the author or any third party may assert apply to any version of the manuscript.

Please use the following link for uploading these materials:
[Redacted]

Best regards,

Aimee Frier
Editorial Assistant
Nature Structural & Molecular Biology
nsmb@us.nature.com

On behalf of

Florian Ullrich, Ph.D.
Associate Editor
Nature Structural & Molecular Biology
ORCID 0000-0002-1153-2040

Reviewer #1:

Remarks to the Author:

The authors have very nicely addressed all my concerns and I recommend it's acceptance.

Reviewer #2:

Remarks to the Author:

Thanks for the new data showing the fusion construct is functional. Since SWELL1 (another name for LRRC8A) is also widely used for VRAC studies in the literature, the authors may consider refer it once in the abstract to increase the cross searchability.

Final Decision Letter:**Message** 15th Nov 2022

:

Dear Raimund,

We are now happy to accept your revised paper "Structure of a volume-regulated heteromeric LRRC8A/C channel" for publication as a Article in Nature Structural & Molecular Biology.

As soon as your article is published, you can generate your shareable link by entering the DOI of your article here: http://authors.springernature.com/share. Corresponding authors will also receive an automated email with the shareable link

Your paper will be published online soon after we receive proof corrections and will appear in print in the next available issue. You can find out your date of online publication by contacting the production team shortly after sending your proof corrections. Content is published online weekly on Mondays and Thursdays, and the embargo is set at 16:00 London time (GMT)/11:00 am US Eastern time (EST) on the day of publication. Now is the

time to inform your Public Relations or Press Office about your paper, as they might be interested in promoting its publication. This will allow them time to prepare an accurate and satisfactory press release. Include your manuscript tracking number (NSMB-A46698B) and our journal name, which they will need when they contact our press office.

About one week before your paper is published online, we shall be distributing a press release to news organizations worldwide, which may very well include details of your work. We are happy for your institution or funding agency to prepare its own press release, but it must mention the embargo date and Nature Structural & Molecular Biology. If you or your Press Office have any enquiries in the meantime, please contact press@nature.com.

Please note that *Nature Structural & Molecular Biology* is a Transformative Journal (TJ). Authors may publish their research with us through the traditional subscription access route or make their paper immediately open access through payment of an article-processing charge (APC). Authors will not be required to make a final decision about access to their article until it has been accepted. [Find out more about Transformative Journals](https://www.springernature.com/gp/open-research/transformative-journals)

Authors may need to take specific actions to achieve [compliance](https://www.springernature.com/gp/open-research/funding/policy-compliance-faqs) with funder and institutional open access mandates. If your research is supported by a funder that requires immediate open access (e.g. according to [Plan S principles](https://www.springernature.com/gp/open-research/plan-s-compliance)) then you should select the gold OA route, and we will direct you to the compliant route where possible. For authors selecting the subscription publication route, the journal's standard licensing terms will need to be accepted, including [self-archiving policies](https://www.springernature.com/gp/open-research/policies/journal-policies). Those licensing terms will supersede any other terms

that the author or any third party may assert apply to any version of the manuscript.

Kind regards,
Florian

Dr Florian Ullrich
Associate Editor, Nature
Consulting Editor, Nature Structural & Molecular Biology
ORCID 0000-0002-1153-2040

Click here if you would like to recommend Nature Structural & Molecular Biology to your librarian:

<http://www.nature.com/subscriptions/recommend.html#forms>